EMBO
Molecular Medicine

# Neuraminidase inhibition promotes the collective migration of neurons and recovery of brain function

Mami Matsumoto [1,2], Katsuyoshi Matsushita[3], Masaya Hane [4], Chentao Wen [5,6], Chihiro Kurematsu[1], Haruko Ota[1,7], Huy Bang Nguyen[8,9], Truc Quynh Thai[8,10], Vicente Herranz-Pérez [11,12], Masato Sawada [1,2], Koichi Fujimoto [3], José Manuel García-Verdugo [11], Koutarou D Kimura [5], Tatsunori Seki [13,14], Chihiro Sato [4], Nobuhiko Ohno[15,16] & Kazunobu Sawamoto [1,2]✉

## Abstract

In the injured brain, new neurons produced from endogenous neural stem cells form chains and migrate to injured areas and contribute to the regeneration of lost neurons. However, this endogenous regenerative capacity of the brain has not yet been leveraged for the treatment of brain injury. Here, we show that in healthy brain chains of migrating new neurons maintain unexpectedly large non-adherent areas between neighboring cells, allowing for efficient migration. In instances of brain injury, neuraminidase reduces polysialic acid levels, which negatively regulates adhesion, leading to increased cell–cell adhesion and reduced migration efficiency. The administration of zanamivir, a neuraminidase inhibitor used for influenza treatment, promotes neuronal migration toward damaged regions, fosters neuronal regeneration, and facilitates functional recovery. Together, these findings shed light on a new mechanism governing efficient neuronal migration in the adult brain under physiological conditions, pinpoint the disruption of this mechanism during brain injury, and propose a promising therapeutic avenue for brain injury through drug repositioning.

**Keywords** Neuronal Migration; Adult Neurogenesis; Chain Migration; Stroke; Drug Repositioning
**Subject Categories** Neuroscience; Pharmacology & Drug Discovery

See also: J Becker & F Szele

## Introduction

There are currently no definitive treatments for brain injury in clinical settings. Although cell transplantation is a regenerative medicine approach that is expected to become a definitive treatment for brain injury, its clinical application has long been a major challenge because of both its invasiveness and its relatively low therapeutic efficacy in large brain areas (Chan et al, 2017; Nakajima et al, 2021). A method to overcome these problems would be a breakthrough in regenerative medicine for neurological diseases, but has not yet been developed. Using the endogenous regenerative capacity of the brain, it would then be possible to develop a minimally invasive method to regenerate cells over wide brain areas.

In the postnatal human brain, endogenous neural stem cells are present until childhood, and in rodents they are present from birth to adulthood; they continuously produce new neurons in limited areas of the brain (Obernier and Alvarez-Buylla, 2019). In the neurogenic ventricular-subventricular zone (V-SVZ) of the lateral ventricles, new neurons form chains to migrate collectively at high speed using each other as scaffolds. Once they reach the olfactory bulb, they integrate into pre-existing neural circuits upon maturation (Lois and Alvarez-Buylla, 1994; Doetsch and Alvarez-Buylla, 1996; Lois et al, 1996). Several cell adhesion molecules, such as N-cadherin and the polysialylated form of neural cell adhesion molecule (PSA-NCAM), are expressed in new neurons and play important roles in chain migration (Yagita et al, 2009; Porlan et al, 2014; Seki and Arai, 1993b; Ono et al, 1994; Chazal et al, 2000), suggesting the importance of cell adhesion in this type of migration. However, it remains unclear how neurons can migrate at high speed in chains that are formed by tightly packed new neurons that adhere to each other.

[1]Department of Developmental and Regenerative Neurobiology, Institute of Brain Science, Nagoya City University Graduate School of Medical Sciences, Nagoya 467-8601, Japan. [2]Division of Neural Development and Regeneration, National Institute for Physiological Sciences, Okazaki 444-8585, Japan. [3]Department of Mathematical and Life Sciences, Hiroshima University, Higashi-Hiroshima 739-8526, Japan. [4]Bioscience and Biotechnology Center, Graduate School of Bioagricultural Sciences, and Institute for Glyco-core Research (iGCORE), Nagoya University, Nagoya 464-8601, Japan. [5]Graduate School of Science, Nagoya City University, Nagoya 467-8501, Japan. [6]Laboratory for Developmental Dynamics, RIKEN Center for Biosystems Dynamics Research, Kobe 650-0047, Japan. [7]Department of Anesthesiology and Intensive Care Medicine, Graduate School of Medical Sciences, Nagoya City University, Nagoya 467-8601, Japan. [8]Section of Electron Microscopy, Supportive Center for Brain Research, National Institute for Physiological Sciences, Okazaki 444-8787, Japan. [9]Department of Anatomy, Faculty of Medicine, University of Medicine and Pharmacy at Ho Chi Minh City (UMP), Ho Chi Minh City 70000, Vietnam. [10]Department of Histology-Embryology-Genetics, Faculty of Basic Medical Sciences, Pham Ngoc Thach University of Medicine, Ho Chi Minh City 70000, Vietnam. [11]Laboratory of Comparative Neurobiology, Cavanilles Institute, University of Valencia, CIBERNED-ISCIII, Valencia 46980, Spain. [12]Department of Cell Biology, Functional Biology and Physical Anthropology, University of Valencia, Burjassot 46100, Spain. [13]Department of Histology and Neuroanatomy, Tokyo Medical University, Tokyo 160-8402, Japan. [14]Department of Anatomy and Life Structure, Juntendo University Graduate School of Medicine, Tokyo 160-8402, Japan. [15]Department of Anatomy, Division of Histology and Cell Biology, Jichi Medical University, Shimotsuke 329-0498, Japan. [16]Division of Ultrastructural Research, National Institute for Physiological Sciences, Okazaki 444-8585, Japan. ✉E-mail: sawamoto@med.nagoya-cu.ac.jp

In the injured brain, new neurons produced from endogenous neural stem cells also form chains and migrate to injured areas, where they then mature (Yamashita et al, 2006; Fujioka et al, 2017). However, brain functions that are lost as a result of injury do not recover spontaneously, likely because of insufficient neuronal migration. Gene transfer to promote the migration of endogenous new neurons, or the implantation of biomaterials to serve as scaffolds for neurons, can promote neuronal migration and neuronal regeneration, thereby recovering brain function (Jinnou et al, 2018; Kaneko et al, 2018; Ohno et al, 2023; Nakajima et al, 2024). However, these surgical approaches have problems such as their invasiveness and localized therapeutic effects (restricted to directly treated areas), which may limit the potential target diseases. To develop treatments that can be applied to many brain diseases, a technique that promotes chain migration in a less invasive and more extensive manner is therefore needed. Although new neurons form chains even in injured brain tissue, their migration efficiency is poor; the mechanisms underlying this poor migration efficiency remain unknown. If the causes of insufficient regeneration are morphologically and molecularly elucidated and a technology can be established that promotes chain migration in the injured brain, it would be a breakthrough in regenerative medicine for neurological diseases.

In the present study, we found that polysialic acid (PSA) forms non-adherent areas in dense, aggregated chains of new neurons to prevent excessive adhesion while they migrate. However, in the injured brain, neuraminidase expression is upregulated, resulting in a decreased PSA, excessive cell–cell adhesion, and reduced migration efficiency. Furthermore, a clinically-used neuraminidase inhibitor maintained PSA in new neurons and allowed them to migrate to injured areas, promoted neuronal regeneration, and recovered brain function. The results of the present study provide insights into the mechanisms of efficient chain migration under physiological conditions as well as those of abnormalities in the injured brain. They also demonstrate the potential applications of these findings to new treatments for brain diseases.

## Results

### New neurons in chains migrating within the RMS have highly restricted adhesion to neighboring cells

To investigate the three-dimensional features of chain-forming new neurons that migrate through the rostral migratory stream (RMS) in the normal brain, we developed a three-dimensional electron microscopic observation method using serial block-face scanning electron microscopy (SBF-SEM), which involves a deep learning-based identification of specific cell types, followed by a semi-automatic segmentation of the 3D cells (Fig. EV1A–B"). The establishment of this method has made it possible to relatively rapidly analyze chain-forming new neurons on a large scale. An extensive examination of the fine morphology of chain-forming new neurons migrating in the normal brain revealed that they maintained non-adherent areas in addition to adherent areas. Although the existence of non-adherent areas between new neurons has been reported (Doetsch et al, 1997), their distribution and significance in the chain have not yet been clarified. In the adult brain, such non-adherent areas between cells, which were not observed outside the neuronal chains (Fig. EV1A), were widely

observed throughout cell chains (Fig. 1A,A', Movie EV1). The non-adherent areas were observed not only in transmission electron microscopy (TEM) and SBF-SEM images after chemical fixation, but also in samples prepared with high-pressure freezing followed by freeze-substitution (Vanhrreveld et al, 1965) (Fig. EV1C–E'). The distance in the non-adherent area between neighboring cells in SBF-SEM was greater than that in TEM and high-pressure freezing, but the proportion of non-adherent area did not differ among all fixation methods (Fig. EV1F,G). These results suggest that the proportion of non-adherent areas was not dependent on the fixation methods. The proportion of non-adherent areas between new neurons inside the chain was higher than those between new neurons and astrocytes surrounding the chain (Fig. 1B–D), suggesting that they may have a particularly important role between new neurons migrating in chains. Interestingly, at the fusion area of chains, non-adherent areas were reduced and adhesion between new neurons was increased (Figs. 1A' arrows,E,F and EV1H–I'). These results suggest that chain-forming new neurons in the normal brain adjust their adhesion to surrounding cells depending on the situation.

### Chain-forming new neurons form excessive cell adhesion in the injured brain

It has been suggested that the efficiency of neuronal migration to injured areas is low relative to the fast chain migration that occurs through the normal RMS (Kaneko et al, 2018; Grade et al, 2013). We therefore compared the morphology of cell chains in the injured brain to those in the normal brain using SBF-SEM. Chains in the normal brain showed an aligned cell orientation and new neurons exhibited smooth morphologies. However, chains in the injured brain had more random cell orientations and new neurons had a shorter leading process (Fig. 2A–E, Movie EV2, 3). Considering a report showing no correlation between neuron morphology and migration speed (Nam et al, 2007), irregular orientation of new neurons does not necessarily indicate low migration speed. However, irregular orientation may prevent new neurons from moving efficiently and effectively to their destination, the site of injury. Brain injury decreased non-adherent area of new neurons at the injured site but not in the RMS (Figs. 2F–H and EV2A). To further characterize the adhesions between new neurons in the injured brain, we observed them at higher magnification, and found small non-adherent areas inside adhesions that we referred to as "pores" (Fig. 2F, G red arrows). The number of small pores—especially those smaller than 0.05 $\mu m^2$—was markedly decreased in new neurons in the injured brain (Figs. 2F–G' red arrows and EV2B–D). These findings suggest that, in the normal brain, in addition to abundant non-adherent areas between migrating new neurons, fine pores are formed within adherent areas to prevent excessive adhesion, thus resulting in relatively fast neuronal migration. In migrating new neurons in the injured brain, both the non-adherent areas and fine pores are reduced, leading to increased adhesion and impaired migration.

### Mathematical modeling suggests efficient chain migration depends on appropriate cell adhesion levels

To investigate whether the levels of adhesion of new neurons affects chain migration, we performed mathematical modeling and simulated

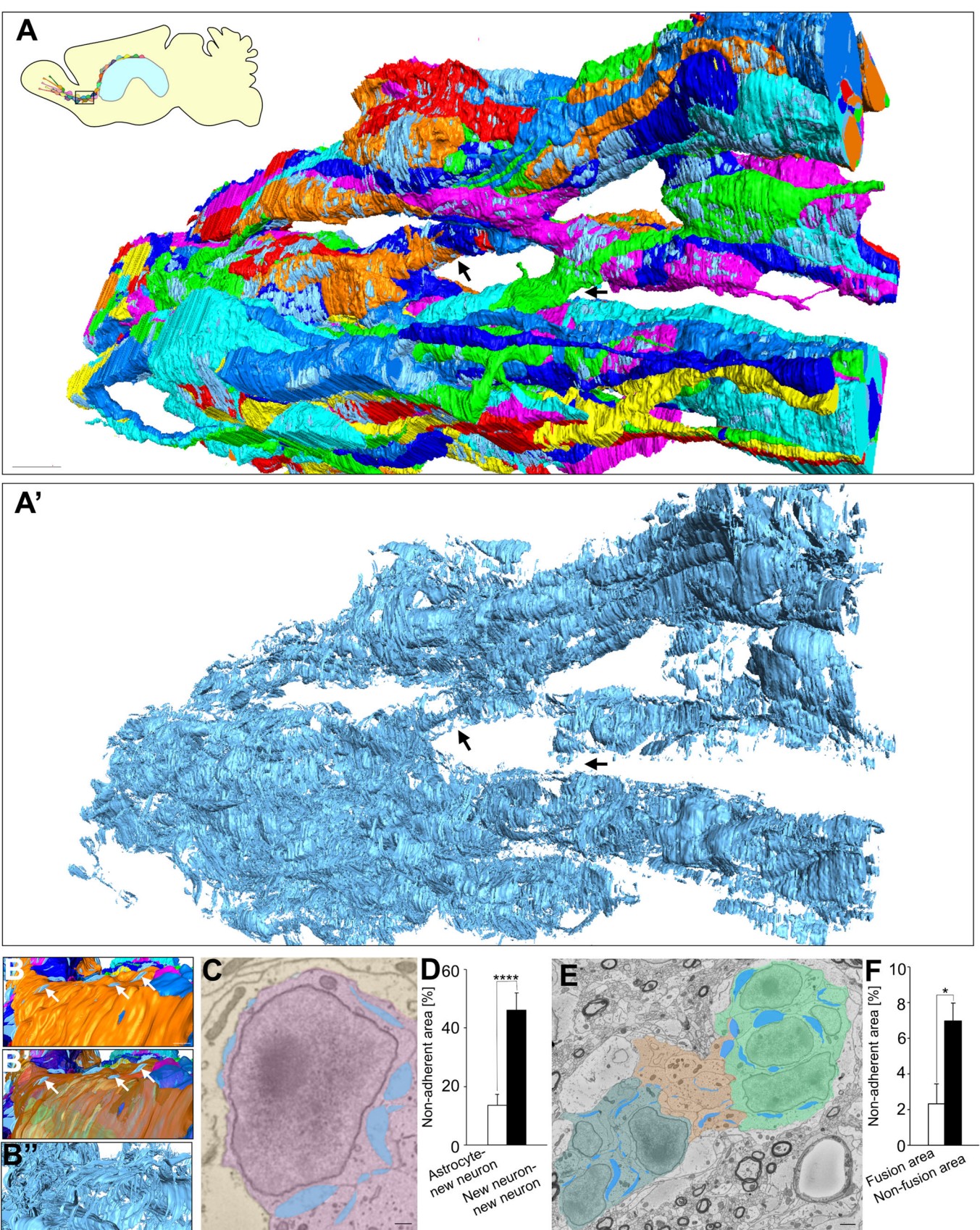

◀ **Figure 1. Non-adherent area is widely observed in neuronal chain.**

(A, A') Three-dimensional re-constructions of neuronal chain (A) and non-adherent area (A') in the adult RMS. The difference in the color of these cells indicate different individual new neurons. Arrows indicate fusion area of chain. In the fusion area, non-adherent area is smaller compared with that in non-fusion area. An interactive 3D model of non-adherent area in chains is shown at https://sketchfab.com/3d-models/non-adherent-area-in-chains-998153cbbd9a432586aa63e4d7e205a7. Scar bar: 5 μm. (B–B") Representative images of three-dimensional re-constructions of new neurons and non-adherent area in chain. White arrows indicate non-adherent area between astrocyte and new neuron. Scar bar: 1 μm. (C) Representative electron microscopic images of new neurons and astrocytes in normal RMS. Yellow, pink, and blue indicate astrocytes, new neurons and non-adherent area, respectively. Scar bar: 1 μm. (D) The proportions of non-adherent area between new neuron and new neuron ($n = 10$ cells), and between astrocyte and new neuron ($n = 13$ cells); $t$-test. (E) Representative electron microscopic images of neuronal chain. Light green, dark green, orange, and blue indicate non-fusion area (chain1), non-fusion area (chain2), chain fusion area, and non-adherent area, respectively. Scar bar: 1 μm. (F) The proportions of non-adherent area in chain fusion area ($n = 3$ areas) and non-fusion area ($n = 4$ areas); $t$-test. Data information: In (D, F), data are presented as mean ± SEM. *$P < 0.05$, ****$P < 0.001$. Source data are available online for this figure.

the neuronal chain migration (Fig. 2I–L, Movie EV4). At the lowest adhesion levels, neurons migrated individually. Increasing adhesion levels resulted in smooth chain migration within a specific range of adhesion levels. Further strengthening of adhesion levels markedly decreased neuronal migration, although neurons were still aggregated. The results of this simulation indicate that efficient chain migration is only possible when appropriate adhesion levels are maintained, and suggest that new neurons migrating to injured areas may have reduced chain migration because of excessive adhesion. We therefore hypothesized that, in addition to cell adhesion molecules, molecules that negatively regulate adhesion may be involved in chain migration.

## New neurons in the injured brain have relatively low PSA expression, thereby increasing cell adhesion

In new neurons migrating in the normal brain, PSA is attached to the adhesion molecule NCAM, which is expressed on the plasma membrane (Fig. EV2E,E'); the presence of PSA is thought to reduce NCAM-mediated cell–cell adhesion (Rutishauser, 2008; Gascon et al, 2010). However, PSA expression levels in migrating new neurons in the injured brain have not yet been clarified. Although the number of PSA-NCAM+ cells was increased by brain injury, PSA signal intensity in each new neuron was lower in the injured brain than in the normal brain (Fig. 2M–P). These results suggest that reduced PSA levels are involved in the increased adhesion between migrating new neurons in the injured brain.

To determine whether the presence of PSA affects the adhesion of new neurons, we administered endoneuraminidaseN (endoN), an enzyme that cleaves PSA, into the normal brain to remove PSA from new neurons. In the endoN-treated group, PSA signal intensity was markedly reduced (Fig. 2Q,R). TEM examination of the fine morphology of cell–cell contacts in these samples revealed that endoN decreased non-adherent areas (closed) (Fig. EV2F–H purple arrows), where adjacent cell membranes were close but not attached (20–35 nm). It also increased adherence-junction (AJ)-like adhesions (Fig. EV2F–H red arrows), the previously reported adhesive structures between neurons (Doetsch et al, 1997). This result suggests that PSA prevents non-adherent areas (closed) from becoming AJ-like adhesions. Similarly, three-dimensional analyses using SBF-SEM revealed decreased non-adherent areas and increased adhesion between neurons in the endoN-treated group (Fig. 2S,T). Additionally, pores were decreased in this group, similar to those between new neurons in the injured brain (Fig. 2S red arrows).

These findings suggest that, in the normal brain, PSA maintains loose adhesion that is appropriate for neuronal migration. In contrast, in the injured brain, decreased PSA in new neurons causes excessive cell adhesion, resulting in migration defects.

## Suppression of neuraminidase expression promotes neuronal migration toward the injured site

A possible exogenous mechanism for the reduced PSA associated with environmental changes is the increased expression of PSA-degrading enzymes, neuraminidases (Neus), in other cells that are present in injured brain tissue as previously reported (Abe et al, 2019). Of the four Neu subtypes, Neu1–4 (Monti et al, 2010), Neu1 and Neu4 reportedly cleave PSA (Sajo et al, 2016; Sumida et al, 2015; Takahashi et al, 2012).

The mRNA expression levels of Neu1–4 in the cortex (Ctx), carpus callosum (CC), and striatum (St) of normal and injured brains were quantified using real-time PCR (Fig. 3A–D). The expression levels of Neu1 (St) and Neu4 (Ctx, CC, and St) were higher in injured brains than in normal brains (Fig. 3A,D). Increased Neu1 and Neu4 expression in the injured brain was also confirmed at the protein level (Fig. EV3A–D'). The increase in Neu1 and Neu4 expression due to brain injury did not reach the V-SVZ or RMS (Fig. EV3A–H). In contrast, Neu2 expression was lower in injured brains than in normal brains, and Neu3 expression was similar between injured and normal brains (Fig. 3B,C). Furthermore, a comparison of Neu1–4 expression levels in the injured brain revealed that Neu1 had the highest expression level, followed by Neu4 (Figs. 3E and EV3I). The decrease in PSA intensity despite the decrease in Neu2 expression by brain injury may be due to the increased expression of Neu1 and Neu4, which are known to be involved in the cleavage of PSA. We therefore focused on Neu1 and Neu4 in subsequent experiments. We used immunohistochemistry to determine which cell types expressed Neu1 and Neu4 in the injured brain. Neu1 was expressed in microglia, astrocytes, and new neurons (Fig. 3F–G") and had a dot-like pattern (Fig. 3F,G), suggesting that it is secreted extracellularly on exosomes in the injured brain, as previously reported in vitro (Sumida et al, 2015). Neu4 was expressed in oligodendrocytes, astrocytes, mature neurons, and new neurons (Fig. EV3J–K"). Together, these results suggest that brain injury increases Neu1 and Neu4 expression in glial cells and new neurons, leading to decreased PSA in new neurons. On the other hand, since Neu2 is reduced by brain injury, Neu2 may be expressed in cells that are reduced in number by injury, such as mature neurons.

Next, to examine the effects of Neu1 and Neu4 on PSA levels and neuronal migration in the injured brain, Neu1 or Neu4 miRNA-expressing viruses were injected into the areas surrounding

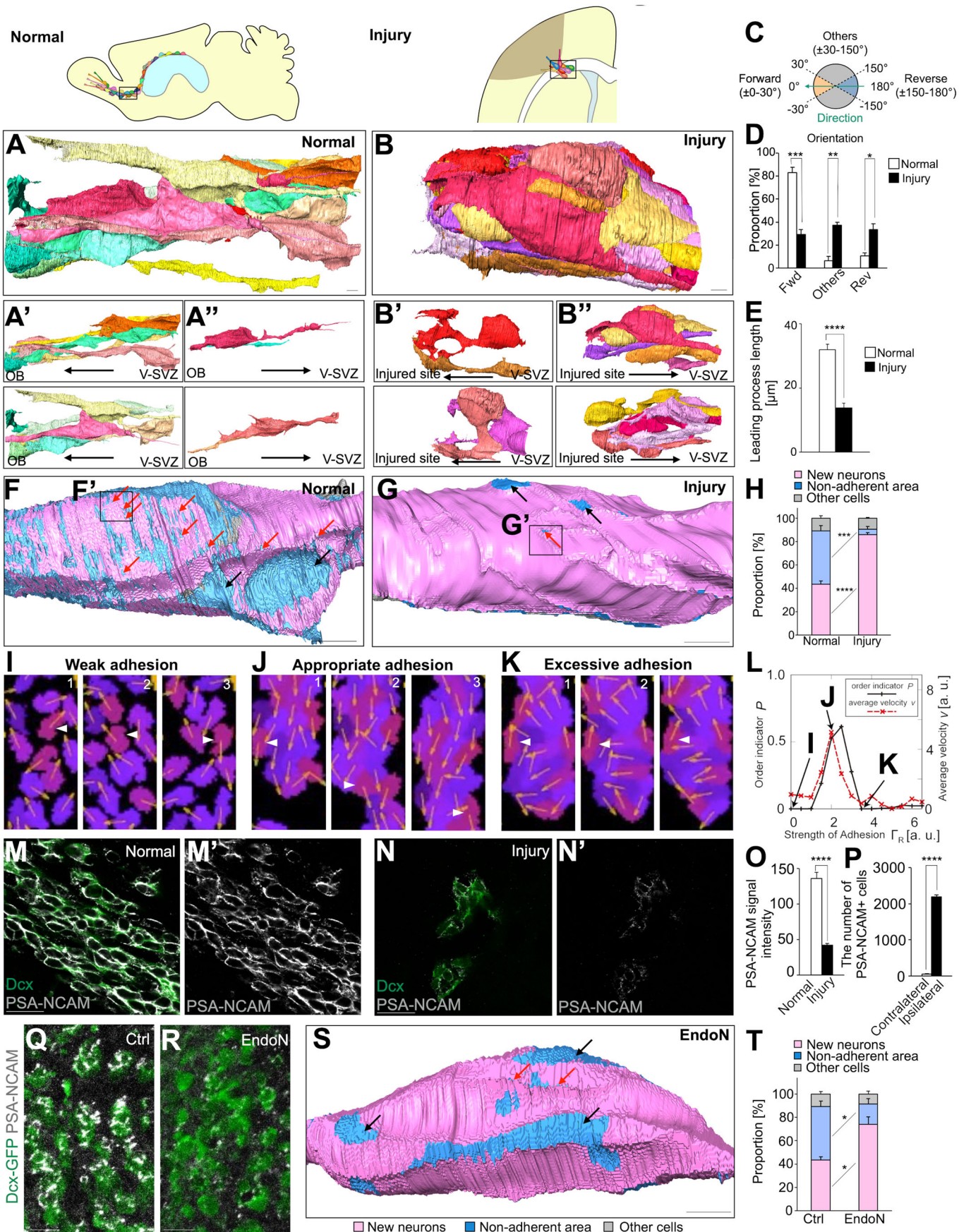

◄

**Figure 2. New neurons migrating toward the injured site show decreased PSA level and increased cell adhesion.**

(A–B") Representative images of three-dimensional re-constructions of neuronal chain in the adult RMS (A) and injured site (B). Images show all new neurons contacting a new neuron indicated in Movie EV2, 3 (A, B). (A') New neurons oriented to the OB. (A") New neuron oriented in the reverse direction. (B') New neurons oriented to the injured site. (B") New neuron oriented in the reverse direction. An interactive 3D model of a neuronal chain in the adult RMS is shown at https://sketchfab.com/3d-models/normal-brain-8961032e439b4290a4bf48be1d267e87. An interactive 3D model of a neuronal chain in the injured site is shown at https://sketchfab.com/3d-models/injured-brain-39702e4e6edf4120843c83394ab11cbb. Scar bars: 3 μm. (C) Mapping of the direction of new neurons. New neurons were classified into forward, reverse and others. (D) Orientation of new neurons in chains in the normal RMS ($n = 3$ chains) and injured site ($n = 3$ chains); $t$-test. (E) The length of neuronal leading process in chains in the normal RMS ($n = 25$ cells) and injured site ($n = 27$ cells); $t$-test. (F–G') The distribution of cell adhesion between new neurons and non-adherent area in neuronal cell body in the adult RMS and injured brain. Pink, blue, and light gray represent adhesion between new neuron and new neurons, non-adherent area, and other cells, respectively. Black and red arrows indicate large space ($\geq 0.05$ μm$^2$) and pore ($<0.05$ μm$^2$), respectively. The boxed areas in (F, G) are enlarged in (F', G' in Figure EV2), respectively. Scar bars: 3 μm. (H) The proportion of non-adherent area in new neurons in the normal RMS ($n = 4$ cells) and injured brain ($n = 3$ cells); $t$-test. (I–K) Simulation snapshots of migrating neurons (purple- or violet-colored regions indicating different individual neurons) in cases with a weak (I), appropriate (J), and excessive strength (K) of cell–cell adhesion. The slight difference in the color of these regions indicates different individual neurons. Black regions represent ECM. The arrows on cells represent the leading process directions $p_m(t)$. Each panel (1, 2, 3) depicts a time course. The white arrowheads focus on the same cell. The unit scales in Fig. 2I–K are arbitrary values. (L) The ordering indicator $P$ (solid black line) and the average velocity $v$ (dashed red line) as a function of the strength of cell–cell adhesion, $\Gamma_R$ (for these strengths, see supplement). The data indicated by (I), (J), and (K) in this panel correspond to panels (I), (J), and (K), respectively. (M–N') Representative image of normal RMS (M, M') and injured corpus callosum (N, N') in WT mice stained for Dcx (green) and PSA-NCAM (white). Scar bars: 20μm. (O) The fluorescence intensity of PSA-NCAM in Dcx+ cells in the normal RMS and in injured brain. PSA levels decreased in migrating Dcx+ cells in the injured brain ($n = 27$ cells) compared with the normal RMS ($n = 30$ cells); Wilcoxon rank sum test. (P) The number of PSA-NCAM+ cells outside of the V-SVZ in the contralateral ($n = 3$ mice) and ipsilateral ($n = 3$ mice) hemispheres; $t$-test. (Q, R) Representative images of RMS in control (Q) and endoN-treated (R) Dcx-GFP mice stained for GFP (green) and PSA-NCAM (white). EndoN removes PSA in the adult RMS. Scar bars: 20 μm. (S) The distribution of cell adhesion in new neurons in control (F) and endoN-treated (S) groups. Black and red arrows indicate large space and pore, respectively. Scar bar: 3 μm. (T) The proportions of non-adherent area in control ($n = 4$ cells) and endoN-treated groups ($n = 3$ cells); $t$-test. Pink, blue, and light gray represent adhesion with new neuron, non-adherent area, and adhesion with other cells, respectively. Data information: In (D, E, H, O, P, T), data are presented as mean ± SEM. *$P < 0.05$, **$P < 0.01$, ***$P < 0.005$, ****$P < 0.001$; adjusted with Bonferroni correction in (D), (H), and (T). Source data are available online for this figure.

injured regions, to suppress the expression of these genes (Fig. 3H). Compared with the control group, in the Neu1 and Neu4 knockdown groups, there was higher PSA signal intensity in new neurons migrating toward the injured area, and more migrating new neurons (Figs. 3I,I',J,J',K,L and EV3L,L'). Increased PSA levels were not observed in any other viral non-infected areas (Figs. 3I",J" and EV3L"). These results indicate that decreased PSA levels in the injured brain are not caused by decreased NCAM expression, but rather by the cleaving of PSA by Neu1 and Neu4. Furthermore, the suppression of Neu1 or Neu4 expression promoted the maintenance of PSA in migrating new neurons, which improved neuronal migration efficiency toward the injured area.

## Neuraminidase inhibitor promotes neuronal migration toward the injured site

To inhibit neuraminidase more extensively in brain tissue, N-Acetyl-2,3-didehydro-2-deoxyneuraminic acid (DANA) or zanamivir was administered for 14 consecutive days, starting at 7 dpi (Fig. 4A). Compared with the control group, the neuraminidase inhibitor group had higher PSA signal intensity in new neurons migrating toward the injured area, as well as improved neuronal migration toward this area (Figs. 4B–F and EV4A–C'). There were no significant effects of inhibitor treatment on cell proliferation or death (Fig. EV4D–F). These results suggest that neuraminidase inhibitor treatment maintains PSA levels in migrating new neurons in the injured brain, and promotes neuronal migration toward the injured site.

We also quantitatively analyzed sialic acid (Neu5Ac) in the RMS and Area1 and Area2, two areas located at different distances from the injury site, using HPLC. Neu5Ac concentrations in Area1 and Area2 of the injured brain were reduced compared to those of normal RMS (Fig. 4G,H). Inhibitor administration increased sialic acid in both of these areas in the injured brain (Fig. 4G,H). These

results suggest that although PSA decreases after injury, neuraminidase inhibitor administration can increase PSA over a wide region in the injured brain.

To determine whether PSA cleavage by microglia-derived neuraminidase impairs neuronal migration, we used co-cultures of new neurons and the microglial cell line BV2 as an in vitro system to mimic the injured brain (Fig. 4I). Because microglia release Neu1 upon the addition of LPS (Sumida et al, 2015), we examined PSA signal intensity and neuronal migration speed after LPS addition. LPS addition decreased PSA signal intensity and the migration speed of new neurons; however, this effect was canceled by the addition of neuraminidase inhibitors (Fig. 4J–S, Movie EV5). These results suggest that neuraminidase cleaves PSA and impairs the migration of new neurons, but that these can be restored by neuraminidase inhibitors.

## Neuraminidase inhibition promotes neuronal regeneration and functional recovery

Fourteen days after inhibitor treatment ended, we analyzed changes in the migration and maturation of neurons that reached the injury site (Fig. 5A). Neuraminidase inhibitor treatment increased the number of NeuN-expressing new neurons (Dcx+NeuN+ cells) that exited the V-SVZ and entered the striatum compared with the control group. Furthermore, although Dcx+NeuN+ cells within 600 μm of the V-SVZ did not differ, Dcx+NeuN+ cells distributed beyond 600 μm of the V-SVZ were increased in the inhibitor-treated group (Fig. 5B–E). Dcx+NeuN+ cells showed a mature morphology with branched and long protrusions (Fig. 5B',C',D' yellow arrows). These results suggest that when the migration of new neurons is promoted by neuraminidase inhibitor treatment, the neurons reach the injured site and mature.

To determine whether inhibitor administration contributes to the recovery of motor function that is lost after injury, foot fault test (FFT)

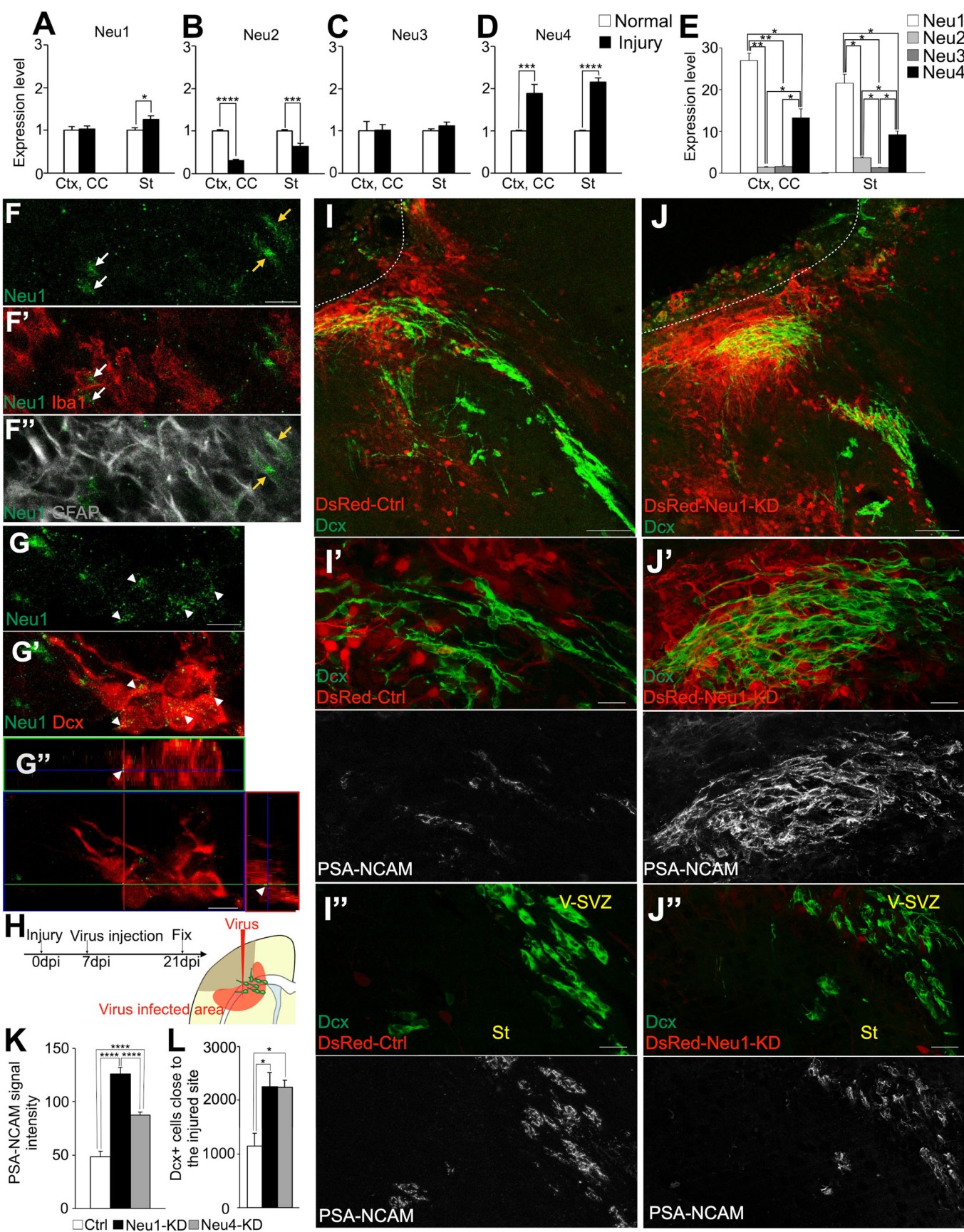

**Figure 3. Neuraminidase suppression in injured brain promoted neuronal migration toward the injured site.**

(A–D) Relative expression levels of Neu1 (**A**) (normal $n = 6$ mice, injury $n = 7$ mice; *t*-test), Neu2 (**B**) (Ctx, CC;normal $n = 6$ mice, injury $n = 6$ mice; *t*-test, St; normal $n = 3$ mice, injury $n = 6$ mice; *t*-test), Neu3 (**C**) (normal $n = 6$ mice, injury $n = 6$ mice; *t*-test, St; normal $n = 3$ mice, injury $n = 6$ mice; *t*-test), and Neu4 (**D**) (Ctx, CC; normal $n = 6$ mice, injury $n = 6$ mice; Wilcoxon rank sum test, St; normal $n = 6$ mice, injury $n = 7$ mice; Wilcoxon rank sum test) in the normal Ctx, CC and St or injured Ctx, CC and St by real-time PCR quantitation. Neu1 and Neu4 expression levels increased after brain injury. (**E**) Comparison of Neu1 to Neu4 expression levels in the injured Ctx, CC and St; Ctx, CC; Neu1; $n = 7$ mice, Neu2 to Neu4; $n = 6$ mice; St; Neu1 to Neu4; $n = 6$ mice; Mann–Whitney U test. (**F–F″**) Representative images of injured brain section in WT mice stained for Neu1 (green), Iba1 (red), and GFAP (white). Merged image of Neu1 and Iba1 in (**F′**) or Neu1 and GFAP in (**F″**). White and yellow arrows indicate Neu1+Iba1+ cells and Neu1+GFAP+ cells, respectively. Scale bar: 10 μm. (**G–G″**) Representative images of injured brain section in WT mice stained for Neu1 (green) and Dcx (red). Merged image of Neu1 and Dcx (**G′**). White arrowheads indicate Neu1+Dcx+ cells. Neu1 dot signals are localized in Dcx+ cell surface (**G″**). Scale bars: 10 μm. (**H**) Experimental scheme. (**I–J″**) Representative images of brain tissues injected with control (**I**) or Neu1-KD (**J**) virus at 7 days post injury (dpi) and stained for Dcx (green) and DsRed (red) at 21 dpi. Representative images of brain tissues injected with control (**I′, I″**) or Neu1-KD (**J′, J″**) virus at 7 dpi and stained for Dcx (green), DsRed (red), and PSA-NCAM (white) at 21 dpi. Dotted lines indicate cortical lesion areas. Scale bars: 100 μm (**I, J**), 20 μm (**I′, I″, J′, J″**). (**K**) The fluorescence intensity of PSA-NCAM in Dcx+ cells close to the injured brain in control ($n = 60$ cells), Neu1 KD ($n = 80$ cells) and Neu4 KD ($n = 61$ cells) groups; Mann–Whitney U test. Neu1 or Neu4 suppression in injured brain maintained the PSA level in Dcx+ cells. (**L**) The number of Dcx+ cells close to the injured site in control ($n = 3$ mice) Neu1 KD ($n = 7$ mice) and Neu4 KD ($n = 5$ mice) groups; Tukey multiple comparisons of means. Neu1 or Neu4 suppression in injured brain caused promotion of neuronal migration toward injured site. Data information: In (**A, B, C, D, E, K, L**), data are presented as mean ± SEM. *$P < 0.05$, **$P < 0.01$, ***$P < 0.005$, ****$P < 0.001$. Source data are available online for this figure.

and elevated body swing test (EBST) were performed immediately before the injury (0 dpi) and at 7 dpi (before inhibitor administration), and 35 dpi (14 days after the end of inhibitor administration) (Fig. 5F). Before the injury, the percentage of left-sided limbs falling out of the net (i.e., percentage of left-sided slips) and the percentage of swings to the right side were approximately 50% in both the control and inhibitor groups. However, 7 days after injury, the percentage of left-sided slips increased and that of right swings decreased in both groups compared with before the injury. At 35 dpi, the percentage of left-sided slips in the inhibitor group decreased and that of right swings in the inhibitor group increased compared with the control group. Moreover, the inhibitor group had a lower percentage of left-sided slips and an increased percentage of right swings at 35 dpi than at 7 dpi (Fig. 5G,H). These results suggest that neuraminidase inhibitor treatment promotes the recovery of motor function lost after brain injury by promoting neuronal migration.

Next, we aimed to determine whether the functional recovery observed in inhibitor-treated animals was caused by the regeneration of V-SVZ-derived neurons. For this purpose, *Nestin-CreERT2; NSE-DTA* mice were administered orallly with tamoxifen eight times (starting 20 days before injury), followed by brain injury; they were then treated with zanamivir for 14 days (starting 7 dpi), and were analyzed 14 days after the end of zanamivir administration (Fig. 5I). In the *Nestin-CreERT2; NSE-DTA* group, Dcx+NeuN+ cells were decreased compared with the *NSE-DTA* group (Fig. 5J). Furthermore, there was no improvement in the percentage of FFT with zanamivir treatment in the *Nestin-CreERT2; NSE-DTA* group (Fig. 5K). These findings indicate that the V-SVZ-derived neurons, regenerated in the injured brain of mice treated with a neuraminidase inhibitor, contribute to functional recovery.

Finally, to determine whether zanamivir administration also promotes neuronal migration in the injured primate brain, common marmoset brains were injured by photothrombosis and treated for 21 days, starting at 7 dpi (Fig. 6A). The number of PSA-NCAM+ cells outside of the V-SVZ was increased by brain injury. However, the PSA signal intensity of these new neurons migrating to the brain injury was lower than that of new neurons migrating through the RMS, but was maintained in zanamivir-treated animals (Fig. 6B–E). Furthermore, zanamivir administration promoted the migration of new neurons toward the injured site (Fig. 6F–H). These results indicate that neuraminidase inhibitor administration promotes neuronal regeneration in primates. Thus, the drug repositioning of zanamivir, an anti-

influenza drug already in clinical use, may lead to new therapies in humans that can promote neuronal migration, neuronal regeneration, and motor function recovery after stroke.

## Discussion

Findings from the present study indicate that migrating new neurons achieve an appropriate level of adhesion in the normal brain by maintaining non-adherent areas via PSA, which allows them to migrate efficiently in the adult brain. In the injured brain, however, PSA in the new neurons is cleaved by neuraminidase, which has elevated expression, thus resulting in excessive adhesion and reduced migration. On the basis of these findings, we propose a novel regenerative medicine: the use of neuraminidase inhibitors to promote neuronal migration and regeneration (Fig. EV5).

How do new neurons migrate in densely-packed chains in the normal brain? We have previously shown that, during chain migration, Rac1 activity creates space between new neurons to allow them to migrate at high speed through densely-packed chains (Hikita et al, 2014). In the current study, chain-forming new neurons migrating in the normal brain had unexpectedly weak adhesion despite their high density. The removal of PSA by endoN treatment resulted in excessive cell adhesion (Fig. 2) and impaired neuronal migration (Johnson et al, 2005), suggesting that weakened adhesion is necessary for neuronal migration. At chain junctions, where there is increased adhesion between new neurons, the migration speed of new neurons may therefore be decreased. Together, these results indicate that new neurons in the normal brain achieve high-speed migration by adjusting intercellular adhesion levels in densely-packed chains.

The removal of PSA by endoN treatment decreased the non-adherent area (closed) and increased AJ-like adhesion in the present study, suggesting that PSA weakens adhesion in the non-adherent area (closed), maintaining a narrow space of 20–35 nm. Intriguingly, this distance is consistent with the intercellular distance that occurs in the presence of PSA-NCAM (Johnson et al, 2005), as predicted from findings using artificial lipid bilayers. EndoN treatment decreases neuronal migration (Ono et al, 1994; Chazal et al, 2000). In the present study, endoN treatment altered non-adherent areas (closed), suggesting that non-adherent areas (closed) are important for neuronal migration. Non-adherent areas

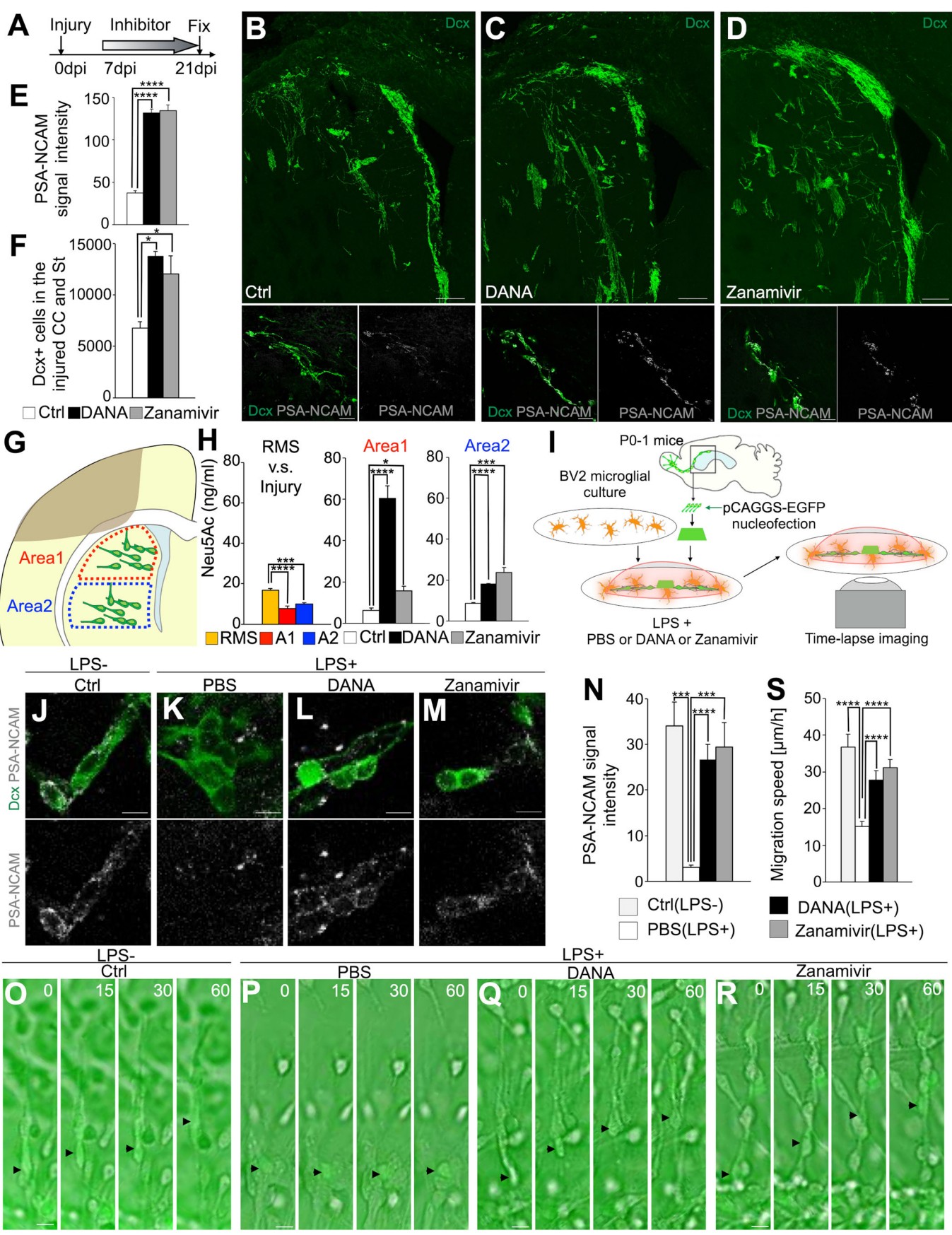

**Figure 4. Neuraminidase inhibitors maintained PSA level and promoted neuronal migration toward the injured site.**

(A) Experimental scheme. (B–D) Representative images of Dcx- and PSA-NCAM-stained section of control (B), DANA (C), and zanamivir (D) administration groups in WT mice. Scale bars: 100 μm, 40 μm(below). (E) The fluorescence intensity of PSA-NCAM in Dcx+ cells in the injured CC and St in control ($n = 60$ cells), DANA ($n = 60$ cells), and zanamivir ($n = 60$ cells) groups; Mann–Whitney U test. DANA or zanamivir administration in brain-injured mice maintained the PSA level in Dcx+ cells. (F) The number of Dcx+ cells in the injured CC and St in control ($n = 3$ mice), DANA ($n = 3$ mice) and zanamivir ($n = 3$ mice) groups; Tukey multiple comparisons of means. DANA or zanamivir administration in brain-injured mice promoted neuronal migration toward injured site. (G) Experimental scheme indicating the brain areas (Area 1 and Area 2) used for the quantification of sialic acid using HPLC. (H) The amount of sialic acid (Neu5Ac) in the RMS and brain areas Area 1 (A1) and Area 2 (A2) indicated in G in PBS, DANA, or zanamivir-treated mice. Left graph, each groups $n = 3$; Tukey multiple comparisons of means. Middle and left graph, PBS $n = 3$ mice; DANA $n = 4$ mice; zanamivir $n = 4$ mice; *t*-test. (I) Experimental scheme. (J–M) PSA-NCAM signal of Dcx+ cells in control (without LPS) (J) and PBS + LPS (K), DANA + LPS (L) and zanamivir+LPS (M) co-cultured with BV2 cells. Scale bars: 10 μm. (N) The fluorescence intensity of PSA-NCAM in Dcx+ cells in control (without LPS) and PBS + LPS, DANA + LPS, and zanamivir+LPS co-cultured with BV2 cells. Control (without LPS) ($n = 6$ cells) and PBS + LPS ($n = 10$ cells), DANA + LPS ($n = 10$ cells) and zanamivir +LPS ($n = 10$ cells); Mann–Whitney U test. PSA-NCAM signal level is reduced by neuraminidase and maintained by neuraminidase inhibitor treatment. (O–R) Time-lapse images of control EGFP+ cells (without LPS) (O), and those treated with PBS + LPS (P), DANA + LPS (Q) and zanamivir+LPS (R) co-cultured with BV2 cells. Numbers indicate minutes after the first imaging frame. Black arrowheads indicate labeled new neurons. Scale bars: 10 μm. (S) Migration speed of new neurons in control EGFP+ cells (without LPS), and those treated with PBS + LPS, DANA + LPS, and zanamivir+LPS co-cultured with BV2 cells. Control (without LPS) ($n = 8$ cells) and PBS + LPS ($n = 30$ cells), DANA + LPS ($n = 30$ cells) and zanamivir+LPS ($n = 30$ cells); Mann–Whitney U test. Data information: In (E, F, H, N, S), data are presented as mean ± SEM. *$P < 0.05$, ***$P < 0.005$, ****$P < 0.001$. Source data are available online for this figure.

(closed) may be an intermediate state that can transition to both non-adherent areas (open) and AJ-like adhesions, thus facilitating rapid and dynamic changes in intercellular adhesion and contributing to chain migration.

The percentage of non-adherent area (open) was not changed by endoN treatment, which suggests that it may be regulated by molecules other than PSA (e.g., Slit, Semaphorin). Considering that tightly packed new neurons in a chain need to dynamically change their morphology to be able to move at high speed (Schaar and McConnell, 2005), non-adherent areas (open) may provide neurons with space for morphological changes, thereby promoting migration.

Chain migration is a form of collective migration (Rørth, 2007). In collective migration other than chain migration, such as that of cancer cells, cells adhere tightly to each other and move together as a cluster (Friedl and Gilmour, 2009). However, chain migration is considered a unique form of collective migration because individual cells move collectively by attaching to and separating from neighboring cells. This form of migration may allow new neurons to flexibly change their destinations on an individual basis. The mechanism identified in the current study, which prevents excessive adhesion by PSA, may allow for this unique mode of migration.

New neurons also form chains in the injured brain, but their migration efficiency is relatively low. In the present study, neuraminidase was increased in the injured brain and PSA was decreased in migrating new neurons, leading to excessive cell adhesion. Why might glial cells release neuraminidase when the brain is damaged? There are known mechanisms in the postnatal mammalian brain that inhibit neuronal regeneration (Yiu and He, 2006), which may help to maintain higher brain functions in mammals by maintaining the precise neural circuits that are constructed during brain development. The suppression of neuronal migration and regeneration by neuraminidase release after brain injury, which was identified in the current study, may be one such function. Our results indicate that neuronal regeneration abilities are not lost in higher animals, but are rather maintained as a hidden ability; regenerative ability was regained simply by removing the inhibition of neuronal regeneration.

We have previously reported that astrocytes act as a physical barrier to neuronal migration in the injured brain (Kaneko et al, 2018). The present study revealed a novel mechanism of

regeneration inhibition: a factor secreted from distant glial cells. This inhibitory mechanism is not caused by glial cells physically inhibiting neuronal migration, but rather by increased cell adhesion between new neurons. The finding that increased cell adhesion between new neurons in the injured brain decreases neuronal migration highlights the importance of mechanisms that negatively regulate cell adhesion during neuronal migration in the adult brain. Although previous therapeutic developments have generally investigated methods to promote the adhesion of migrating new neurons to scaffolds in the injured brain (Jinnou et al, 2018; Ohno et al, 2023), we used a completely different therapeutic approach. By targeting molecules that negatively regulate the adhesion of migrating new neurons, we were able to confer high migratory efficiency, similar to that of new neurons in the normal brain.

Surgical treatment methods, such as gene delivery or supplying artificial migratory scaffolds to migrating new neurons (Nakajima et al, 2021), have limited therapeutic effects in areas around the administration site, thus making it difficult to regenerate neurons in entire injured areas of human brains, which have relatively large volumes (Paredes et al, 2016). A previously reported treatment that involves injecting a virus vector expressing PST, an enzyme that synthesizes PSA, into the injured brain has similar limitations (El Maarouf et al, 2006). The idea of injecting a drug that removes PSA into the brain and using the dispersed neurons from the V-SVZ for treatment has also been reported (Battista and Rutishauser, 2010), but this method would be invasive and would causes loss of PSA even in normal neurons. The new treatment, reported here, is advantageous to previously reported therapies because it is minimally invasive and can promote extensive neuronal migration via drug administration. In addition, there may be advantages in administrating drugs whose duration can be limited to the period of neuronal migration, because permanent intervention may continue to increase PSA even after neuronal migration ends, thus causing unexpected adverse effects.

Of the neuraminidase inhibitors used as anti-influenza drugs, zanamivir has less inhibitory activity against human neuraminidase than against viral neuraminidase (Hata et al, 2008); however, it may inhibit human neuraminidase at relatively high doses. Because our study showed that the inhibition of either Neu1 or Neu4 promoted neuronal migration, it is possible that the

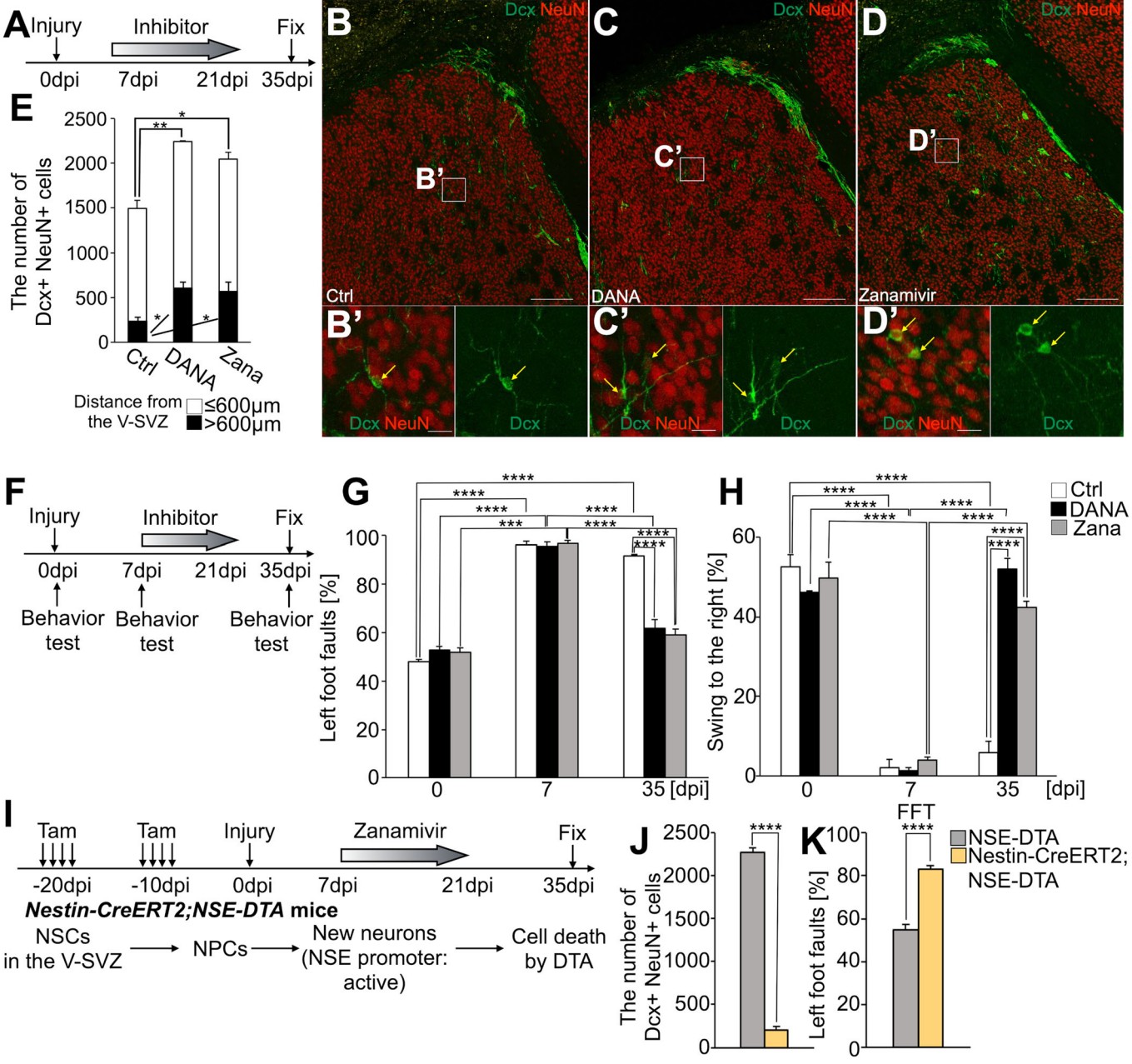

**Figure 5. Neuraminidase inhibitors administration promoted neuronal regeneration and brain functional recovery after brain injury.**

(A) Experimental scheme. (B–D') Representative images of brain section in control (B), DANA (C), and zanamivir (D) administration groups in WT mice stained for Dcx (green) and NeuN (red). The boxed areas in (B, C, D) are enlarged in (B', C', D'), respectively. Yellow arrows indicate Dcx+NeuN+ cells. Scale bars: 200 μm (B, C, D), 20 μm (B', C', D'). (E) The number of Dcx+NeuN+ cells in the injured CC and St in control, DANA, and zanamivir groups; each group $n = 3$ mice; Tukey multiple comparisons of means. DANA or zanamivir administration in injured brain increased neuronal maturation in the injured brain. (F) Experimental scheme. (G) The percentage of left foot faults in the foot fault test (FFT) in control, DANA, and zanamivir groups; each group $n = 3$ mice; Tukey multiple comparisons of means. (H) The percentage of swing to the right in the elevated body swing test (EBST) in control, DANA, and zanamivir groups; each group $n = 3$ mice; Tukey multiple comparisons of means. (I) Experimental scheme. (J) The number of Dcx+NeuN+ cells in *NSE-DTA* and *Nestin-CreERT2; NSE-DTA* mice administered with zanamivir; each group $n = 3$ mice; t-test. (K) The percentage of left foot faults in *NSE-DTA* and *Nestin-CreERT2; NSE-DTA* mice administered with zanamivir; each group $n = 3$ mice; t-test. Data information: In (E, G, H, J, K), data are presented as mean ± SEM. *$P < 0.05$, **$P < 0.01$, ****$P < 0.001$. Source data are available online for this figure.

inhibition of either Neu1 or Neu4, or both, may have a therapeutic effect. Furthermore, by demonstrating the therapeutic effects of zanamivir in mice and non-human primates, our findings indicate the very real possibility of using drug repositioning to develop neuroregenerative therapies.

In conclusion, our study has revealed a novel regulatory mechanism of neuronal migration under physiological conditions. By applying this regulatory mechanism under pathological conditions, we have proposed an innovative treatment for brain injury that was previously considered impossible.

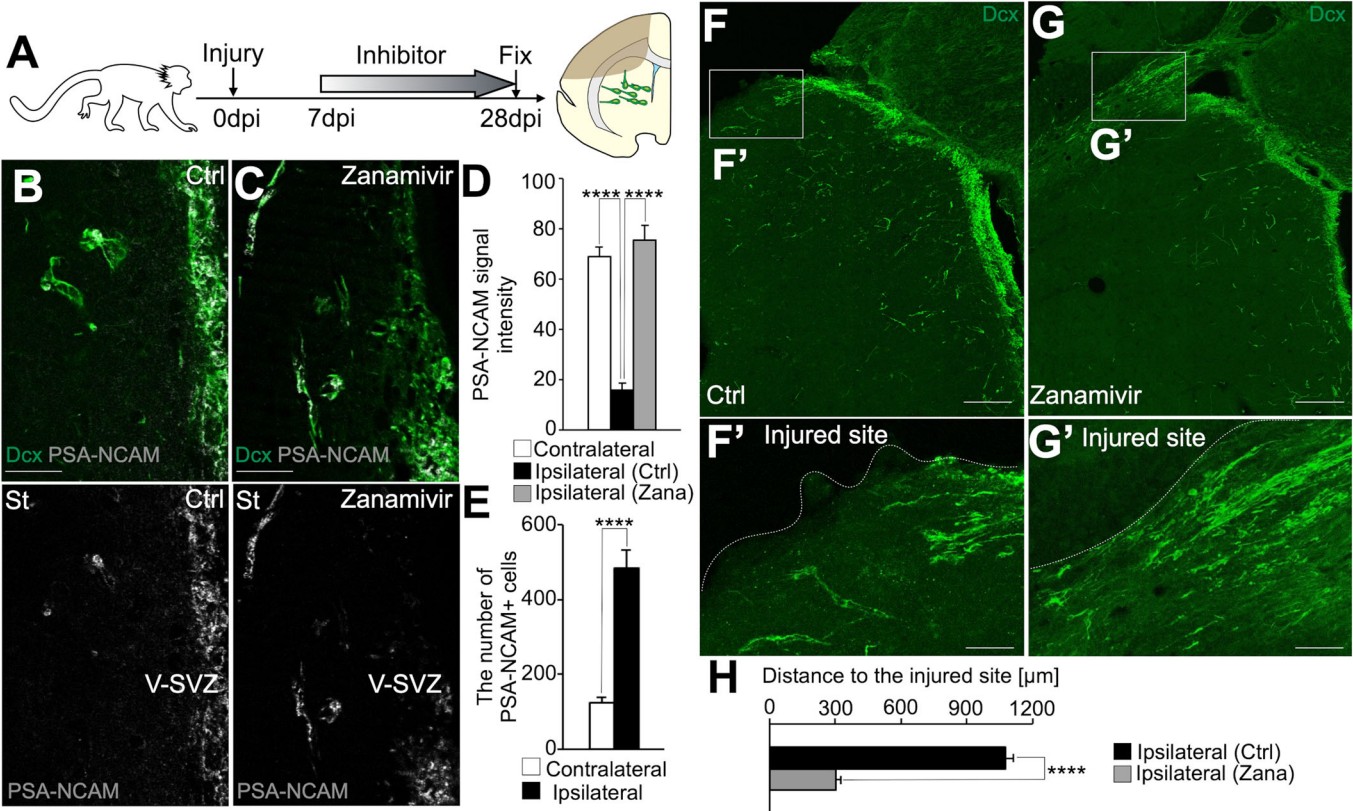

**Figure 6. Neuraminidase inhibitor administration promoted neuronal migration in the primate injured brain.**

(A) Experimental scheme. (B, C) Representative images of common marmoset brain sections stained for Dcx (green) and PSA-NCAM (white) in control (B) and zanamivir (C) administration groups. Scale bars: 50 μm. (D) The fluorescence intensity of PSA-NCAM in Dcx+ cells in the contralateral RMS (contralateral), and the brain areas outside of the V-SVZ in the ipsilateral hemisphere in the control (Ctrl) and zanamivir (Zana) groups; Contralateral RMS, $n = 15$ cells, Control, $n = 30$ cells; Zanamivir, $n = 30$ cells; Mann–Whitney U test. The zanamivir administration maintained the PSA level in new neurons in injured brain. (E) The number of PSA-NCAM+ cells outside of the V-SVZ in the contralateral ($n = 3$ marmosets) and ipsilateral ($n = 3$ marmosets) hemispheres; $t$-test. (F–G′) Representative images of common marmoset brain sections stained for Dcx (green) in control (F, F′) and zanamivir (G, G′) administration groups. The boxed areas in (F and G) are enlarged in (F′ and G′), respectively. Scale bars: 200 μm (F, G), 50 μm (F′, G′). (H) The Distance of Dcx+ cells to the injured site in the injured brain in control and zanamivir groups. Zanamivir administration promoted neuronal migration in the injured brain. Control, $n = 228$ cells; Zanamivir, $n = 418$ cells; Wilcoxon rank sum test. Data information: In (D, E, H), data are presented as mean ± SEM. ****$P < 0.001$. Source data are available online for this figure.

# Methods

## Animals

Wild-type (WT) male and female C57BL/6J mice were purchased from Japan SLC (Shizuoka, Japan) (RRID: IMSR_JAX:000664). *Dcx-EGFP* mice (Gong et al, 2003) were provided by the Mutant Mouse Research Resource Center (MMRRC). *Nestin-CreERT2;-neuron-specific enolase(NSE)-diphtheria toxin fragment A(DTA)* mice were previously described (Imayoshi et al, 2008). Common marmosets (*Callithrix jacchus*) obtained from mating pairs in a domestic animal colony were used. Tamoxifen administration was performed as described previously (Imayoshi et al, 2008). All experiments involving live animals were performed in accordance with the guidelines and regulations of Nagoya City University.

## Immunohistochemistry and image acquisition

Immunohistochemistry in mouse brain sections was performed as described previously (Sawada et al, 2018; Matsumoto et al, 2019).

Briefly, the adult brains were fixed by transcardiac perfusion with 4% paraformaldehyde (PFA) in 0.1 M phosphate buffer (PB), and postfixed overnight at 4 °C in the same fixative. Fifty-micrometer-thick floating coronal and sagittal sections were made using a vibratome (VT-1200S, Leica), and incubated for 30 min at room temperature (RT) in 10% normal donkey serum (NDS) and 0.2% Triton X-100 in PBS (blocking solution). These sections were incubated with primary antibodies in blocking solution overnight at 4 °C, and with AlexaFluor488/568/647-conjugated secondary antibodies (1:1000, Invitrogen) for 2 h at RT in blocking solution. The following primary antibodies were used: rabbit anti-doublecortin (Dcx) (1:1000, 4604S, Cell Signaling Technology; RRID:AB_10693771); guinea pig anti-Dcx (1:400, AB2253, Millipore; RRID:AB_1586992); rabbit anti-DsRed (1:1000, 632496, Clontech; RRID:AB_10013483); mouse IgM anti-PSA-NCAM (1:1000, 12E3) (Seki and Arai, 1991); rabbit anti-Neu1 (1:50, ab244386; Abcam); rabbit anti-Neu4 (1:100, 12995-1-AP; Proteintech; RRID:AB_2149206); goat anti-Iba1 (1:1000, ab5076; Abcam); mouse IgG anti-GFAP (1:1000, G3893; Sigma-Aldrich; RRID:AB_477010); rabbit anti-Olig2 (1:100, JP18953; IBL;

RRID:AB_1630817); rabbit anti-NeuN (1:1000, ab177487; Abcam; RRID:AB_2532109); mouse anti-Mash1 (1:500, sc-374104; Santa Cruz Biotechnology; RRID:AB_10918561); rabbit anti-Ki67 (1:500, ab16667; abcam; RRID:AB_302459); guinea pig anti-Neurofilament L (NF) (1:200, 171 014; Synaptic Systems; RRID:AB_2864783) antibodies. Immunohistochemistry in common marmoset brain sections was performed as described previously (Akter et al, 2020). Briefly, the adult brains were fixed by transcardiac perfusion with 4% paraformaldehyde (PFA) in 0.1 M phosphate buffer (PB), and postfixed overnight at 4 °C in the same fixative. Sixty-micrometer-thick floating coronal and sagittal sections were made using a vibratome (VT-1200S, Leica), and incubated for 1 h at room temperature (RT) in blocking solution. These sections were incubated with primary antibodies in blocking solution overnight at 4 °C, and with AlexaFluor488/647-conjugated secondary antibodies (1:1000, Invitrogen) for 3 h at RT in blocking solution. The following primary antibodies were used: guinea pig anti-Dcx (1:400, AB2253, Millipore; RRID:AB_1586992); mouse IgM anti-PSA-NCAM (1:400) (Seki and Arai, 1991) antibodies.

Images were acquired by scanning at 2-µm intervals using an LSM 700 confocal laser-scanning microscope (Carl Zeiss) with a 20× objective lens. The fluorescence intensity of PSA-NCAM in the Dcx+ cells was quantified using ZEN software (Carl Zeiss). The Dcx+ cells or Dcx+NeuN+ cells in the injured brain in every sixth 50-µm-thick coronal section were counted, and then the total number per hemisphere was estimated by multiplying the sum of the counted cells by six.

## Serial block-face scanning electron microscopy (SBF-SEM) and cell reconstruction

Sample preparation, observation with SBF-SEM and analyses of acquired data were performed as described previously with slight modifications (Matsumoto et al, 2019). SBF-SEM observation of the normal RMS and injured lesion was performed using a Merlin or Sigma scanning electron microscope (Carl Zeiss) equipped with a 3View in-chamber ultramicrotome system (Gatan). Serial image sequences were 40.96 × 40.96 µm wide (5.0 nm/pixel) and >80 µm deep with 80-nm steps. Sequential images were processed using FIJI. Manual segmentation of the cell membrane and non-adherent areas was performed using Microscopy Image Browser (Belevich et al, 2016). A semi-automatic segmentation was performed using Dragonfly (Object Research Systems) in conjunction with Seg2Link (Wen et al, 2023). The two-types of training datasets were generated using Dragonfly by manually annotating the cytoplasm, cell membrane, and space, or the cytoplasm, cell membrane, and space of neurons, and non-neuron cells, in a small portion randomly selected from the entire EM image. These created annotations were used to train the deep neural networks with the Sensor-3D architecture within Dragonfly. Once trained, Dragonfly automatically predicted the cytoplasm areas and the new neuron areas in the entire set of EM images. These predictions were then imported into Seg2Link to generate instance segmentation of the new neurons. This process involved automatic segmentation and manual corrections to divide the attached neuron areas into individual ones, as previously described (Wen et al, 2023) with additional processing of the neuron boundaries to ensure the accurate detection of non-adherent areas and cell adhesions. Three-dimensional reconstruction of migratory new neurons and

their non-adherent area was performed using Amira software (Thermo Fisher Scientific) as reported previously (Matsumoto et al, 2019).

## Photothrombotic injury

To induce cortical injury, we used a modified version of the photothrombosis technique described previously (Watson et al, 1985). Mice (8 to 12 weeks old) were anesthetized with an oxygen/isoflurane mixture (97.5/2.5%) administered through an inhalation mask and placed on a 37 °C heating bed. The scalp was incised to expose the skull surface and cleaned to reveal the bregma and the target area for illumination. For illumination, a fiber optic cable delivering a light source 8 mm in diameter (MSG6-1100S, Moritex, Saitama, Japan) was placed stereotaxically onto the skull 1 mm anterior and 2.7 mm lateral from the bregma. Cortical injury was induced by activation of photosensitive rose bengal dye (30 mg/kg, 330000-1G; Sigma-Aldrich, St. Louis, MO, USA) in PBS injected intravenously. The skull was illuminated for 10 min (533 nm; MHAA-100W-100V, Moritex). For common marmosets, we used a modified version of the photothrombosis technique described previously (Ikeda et al, 2013). Cerebral ischemia in common marmosets (3 months old) was produced via intravascular thrombosis induced by an intravenous injection of rose bengal (30 mg/kg) and irradiation with green light (533 nm, metal halide lamp, PCS-MH357RC, Nippon P.I Co., Ltd., Tokyo, Japan) for 5–10 min under deep anesthesia. All surgeries were performed under general anesthesia induced by intramuscular injection of medetomidine (0.04 mg/kg), midazolam (0.4 mg/kg) and butorphanol (0.4 mg/kg), maintained by oxygen/isoflurane mixture (97.5/2.5%) and antagonized by atipamezole (0.15 mg/kg). The scalp was incised to expose the skull surface and cleaned to reveal the bregma and the target area for illumination. For illumination, a fiber optic cable delivering a light source 12 mm in diameter (PLG-1-1000-3R-UX350, Nippon P.I. Co., Ltd., Tokyo, Japan) was placed stereotaxically onto the skull 5 mm anterior, 6 mm lateral to the bregma. Only the individuals with an infarction area covering 20 to 30% of the ipsilateral hemisphere were used for analysis.

## Transmission electron microscopy

The sample preparation for transmission electron microscopy (TEM) was performed as described previously (Matsumoto et al, 2019). Adult male mouse brains were fixed by transcardiac perfusion with 2% PFA and 2.5% GA in 0.1 M PB (pH 7.4) at 4 °C, and postfixed overnight at 4 °C in the same fixative. These brains were cut into 200-µm-thick coronal sections using a vibratome (VT-1200S, Leica) (anterior +2.0–2.3 mm from the bregma). The sections were treated with 2% osmium tetroxide (OsO$_4$, Electron Microscopy Sciences) in 0.1 M PB (pH 7.4) at 4 °C, dehydrated with a graded series of ethanol, placed in propyleneoxide (Nisshin EM), and embedded in Durcupan resin for 72 h at 60 °C. Semi-thin sections (1.5-µm-thick) were sequentially cut using an ultramicrotome (UC6, Leica) with a diamond knife (histo, DiATOME), and stained with 1% toluidine blue, and sections of interest were chosen under a light microscope CX23 (Olympus). Ultra-thin sections (60–70-nm-thick) were sequentially cut from the embedded semi-thin sections using an ultramicrotome (UC6, Leica) with a diamond knife (SYM2045,

SYNTEK), and stained with 2% uranyl acetate in distilled water for 15 min and with modified Sato's lead solution for 5 min. TEM images were acquired using a JEM-1400plus (JEOL) fitted with a digital camera.

Immunoelectron microscopy for PSA-NCAM was performed as previously described (Seki and Arai, 1993a). Adult male Wistar rats were perfused with a mixture of 4% PFA and 0.1% GA in PB at 4 °C under deep sodium pentobarbital anesthesia. The brains were then removed and immersed overnight in 4% PFA in PB at 4 °C. After washing with PBS, vibratome sections were cut at 50 µm and rinsed with PBS. Preembedding immunostaining was performed according to a method described (Martini and Schachner, 1986) with minor modifications. In brief, these vibratome sections were incubated in 0.1 M $NaIO_4$ (10 min) and then in $NaBH_4$ (15 min), followed by immersion in 5% dimethyl sulfoxide (30 min) at room temperature. Next, the sections were incubated with mouse IgM mAb 12E3 (1:5000) (Seki and Arai, 1991) overnight at 4 °C and then with anti-mouse IgM conjugated with peroxidase (1:100) for 3 h at room temperature, and fixed with 1% GA for 1 min at 4 °C. Following this, the sections were incubated with DAB solution for 30 min and then with a DAB solution containing 0.005% $H_2O_2$ for 5–10 min. Each of the above steps was followed by washing with PBS. Finally, the sections were postfixed with 2% $OsO_4$ in PB, dehydrated, and embedded in Epok 812 (Oken-Shoji Co., Ltd., Tokyo).

## High-pressure frozen and freeze substitution

Fresh adult male mouse brains were cut into 200-µm-thick coronal sections using a vibratome (VT-1200S, Leica). The sections were high pressure frozen using EM PACT (Leica). Freeze substitution of frozen sections was carried out using the EM AFS system (Leica) with 1% osmium tetroxide in acetone at –80 °C for 92 h. The temperature was increased by 10 °C/h and held at –20 °C for 19 h. Finally, the samples were warmed at 4 °C for 6 h. These samples were washed five times with cold acetone, placed in propylene-oxide, and embedded in Durcupan resin for 72 h at 65 °C. Ultra-thin sections of the embedded specimens were prepared and observed as described above.

## Co-culture of V-SVZ-derived new neurons with BV2 cells

In vitro culture and time-lapse imaging of V-SVZ-derived new neurons were performed as previously described (Matsumoto et al, 2019). V-SVZ tissues from P0–P3 male and female WT mice were dissected and dissociated with trypsin-EDTA (Invitrogen). The cells were washed with L-15 medium (Invitrogen) containing 40 µg/ml DNase I (Roche Diagnostics) and transfected with the plasmid pCAGGS-EGFP using the Amaxa Nucleofector IV system (Lonza Walkersville). Transfected cells were collected in RPMI-1640 medium (Thermo Fisher Scientific) and reaggregated. These cell aggregates were then placed on a monolayer of BV2 cells (provided by Dr. Takatoshi Ueki, Nagoya City University) embedded in 60% Matrigel (BD Biosciences) and grown in Neurobasal medium containing 2% NeuroBrew-21 (Invitrogen), 2 mM L-glutamine (Invitrogen), and 50 U/ml penicillin-streptomycin (Invitrogen) for 2 days for time-lapse imaging. Treatment of cells with lipopoly-saccharide (LPS; 100 ng/ml$^{-1}$ µg/ml), with or without a neurami-nidase inhibitor (DANA or Zanamivir; final 1 mM), was performed

as previously described (Sumida et al, 2015). Time-lapse imaging of randomly selected EGFP-expressing new neurons was performed using a fluorescence microscope (KEYENCE BZ-X810) and ×20 objective lens.

## Real-time PCR

To examine the mRNA levels of Neu1–4 in the normal and injured brain, the brain of normal or 7 dpi WT mice under deep anesthesia was rapidly removed from the skull, and the Ctx, CC, and St were dissected. Total RNA was extracted from the collected samples using a CellAmp Direct RNA prep kit (Takara). To examine the mRNA levels of Neu1–4 in normal Ctx, CC, and St or injured Ctx, CC, and St, the total RNA was extracted from the tissue with TRIzol reagent (Invitrogen). cDNA was synthesized using Super-Script IV Reverse Transcriptase (Invitrogen). Quantitative SYBR Green real-time PCR was carried out as previously described (Zheng et al, 2015), with an ABI 7500 Fast Real-Time PCR instrument (Applied Biosystems, Foster City, CA, USA). The data are expressed as the fold change in gene expression relative to the normal brain group. The following primers were used: 5'-TGCCAGCCCTACGAGCTT-3' and 5'-TGGTTCCGGGCGTT GAT-3', which amplified a 52-bp Neu1 product; 5'-CCCTGGG CGTGTATCAGAA-3'; 5'-AGCACAGCCGTGTGACATTAAC-3', which amplified a 61-bp Neu2 product; 5'-GGCTGGGAC GGCTGGTA-3'; 5'-TCGAAATCGGCTTGGTGTTC-3', which amplified a 60-bp Neu3 product; 5'-AAGGCACGTCCTTCC-TACCA-3'; 5'-GCAACCACGAGCCGTCTCT-3', which amplified a 61-bp Neu4 product; and 5'-CATGGCCTTCCGTGTTCCTA-3' and 5'-CACGTCAGATCCA-3', which amplified a 55-bp GAPDH product.

## Viral vectors and plasmids

The generation of Neu1- and Neu4-knockdown (KD) plasmids was performed as described previously (Ota et al, 2014; Sawada et al, 2018). Briefly, the target sequence of the mouse Neu1, Neu4, or lacZ (control) gene was inserted into a modified Block-iT Pol II miR RNAi expression vector containing DsRed-Express. The Gateway system (Invitrogen) was used to generate the following CSII lentiviral expression vectors: CSII-EF-DsRed-Express-miR-Neu1, -Neu4, or -LacZ miRNA. The KD efficiency of these vectors was assessed by western blotting as described previously (Ota et al, 2014). To generate lentiviral particles, lentiviral vectors and packaging vectors (pCAG-HIVgp and pCMV-VSV-G-RSV-Rev) were transfected into HEK293T cells. Three days after transfection, the culture supernatants were concentrated by centrifuging at 8000 rpm at 4 °C for 16 h using an MX-307 refrigerated micro-centrifuge (Tomy). A 2-µL volume of lentiviral suspension was stereotaxically injected into the surrounding injured area (1.8, 1.6, 1.4, 1.2, 1.0, 0.8, 0.5 mm anterior, 1.2 mm lateral to lambda and 1.5–2.0 mm deep) at 7 dpi.

## EndoN treatment and neuraminidase inhibitor administration

One µl of PBS or endoN (10 units/µl) was stereotaxically injected into the lateral ventricle (−0.2 mm anterior, 0.6 mm lateral, 2.0 mm

deep from bregma). For inhibition of neuraminidase, DANA (50 mg/kg), zanamivir (50 mg/kg), or PBS (control) was injected intraperitoneally every day on 7–21 dpi in mice and on 7–28 dpi in common marmosets.

## Sialic acid quantification

The dissected mouse brain samples were homogenized with PBS containing 1% Triton X-100, protease inhibitor cocktail (1 μg/mL aprotinin, 1 μg/mL leupeptin, 1 μg/mL pepstatin, and 2 μg/mL antipain), and 5 mM ethylenediaminetetraacetic acid (EDTA), and incubated on ice for 1 h. Homogenates were centrifuged, and the protein concentration of the supernatants was evaluated by the bicinchoninic acid (BCA) assay. The sialic acid (Sia) in each sample was quantified using the DMB derivatization method as previously described (Sato et al, 1999). Briefly, 1 μl of brain homogenate was hydrolyzed in 0.2 M trifluoroacetic acid (TFA) by incubating at 80 °C for 2 h. The sample was dried in a Concentrator plus (Eppendorf, Germany). For DMB derivatization, 20 μl of 0.01 M TFA and 20 μl of DMB solution were added to the dried samples, following by incubating at 50 °C for 2 h. For Fluorometric HPLC analysis, the sample was diluted 10-fold and injected through a JASCO intelligent inert sampler (AS-4050) into a Handy ODS column (250 × 4.6 mm i.d.) (Wako, Japan) and eluted with methanol/acetonitrile/water (7/9/84, v/v/v). The HPLC system consisted of a JASCO pump (PU-4180), JASCO fluorescence detector (FP-2025 plus, excitation, 373 nm; emission, 448 nm), and Chromato-PRO Integrator (Run Time Corporation, Japan).

## Behavior tests

Wild-type, *Nestin-CreERT2;NSE-DTA* and *NSE-DTA* mice were repeatedly subjected to quantitative neurological testing on the pre injury, first, and 5th week after injury. The foot-fault test was performed as previously described (Hernandez and Schallert, 1988) with modifications. Briefly, mice were placed on an elevated wire hexagonal grid with 40-mm wide openings, and allowed to roam freely. A misstep was recorded as a foot fault when the mouse slipped or fell with one of its limbs dropping into an opening in the grid. The number of foot-faults for each limb was separately counted for 10 min, and then the ratio of the number of contralateral (left) fore- and hindlimb faults to the total number for the four limbs was calculated as a percentage. The elevated body swing test was performed as previously described (Borlongan and Sanberg, 1995) with modifications. The mice were held at 1.5 cm from the base of their tail, then elevated to approximately 30 cm above the surface. A swing was recorded when the mouse moved its head more than 90° from the vertical axis to either side for 90 s. The percentage of the number of right-side swings relative to the total number of swings was calculated.

## Experimental design and statistical analysis

Statistical analysis was performed using EZR (Kanda, 2013) as previously reported (Kaneko et al, 2018). At least three independent experiments were performed for each quantification. All data were expressed as the mean ± standard error of the mean. For the analyses of neuronal maturation (Fig. 5E) and cell proliferation or death (Fig. EV4D–F) investigators were blinded during

quantification. Normality and equal variances between group samples were assessed using the Shapiro-Wilk test and $F$ test, respectively. When normality and equal variance between sample groups were achieved, the significance of differences between means was determined by a two-tailed Student's $t$ test, or one-way ANOVA, followed by Tukey test. Where the normality or equal variance of samples failed, a Mann–Whitney $U$ test or Wilcoxon signed-rank test followed by Bonferroni test was performed. A $P$ value of <0.05 was considered to be statistically significant.

## Mathematical model of neuronal migration

To examine the effects of cell–cell adhesion on collective migration, we calculated the ordering indicator $P$ of the leading process direction of model cell (Vicsek et al, 1995). This indicator reflects the order of the migrating direction of cells and indicates the emergence of collective migration such as chain migration. The value of the indicator $P$ is unity when the directions are perfectly ordered and is zero when the directions are uniformly random. Here, for convenience regarding the definition of $P$, we denote the leading process direction of the $m$th cell by $\boldsymbol{p}_m(t)$ at time $t$, In this case, the definition of $P$ is

$$P = \left| \frac{1}{T} \int_T \frac{1}{N} \sum_m \boldsymbol{p}_m(t) \right|. \tag{1}$$

Here, $t$ is time, and $T$ is the average period over time. $N$ is the number of cells. The summation over $m$ is taken over all cells for the calculation of the average value over cells. $T$ is set to $5 \times 10^3$ cycles of typical saltatory migration. Before this calculation, we obtain the steady state through a sufficiently long simulation with $10^3$ cycles of typical saltatory migration.

We also calculate the average velocity over cells,

$$v = \left| \frac{1}{T} \int_T \frac{1}{N} \sum_m \boldsymbol{d}_m(t) \right|. \tag{2}$$

Here, $\boldsymbol{d}_m(t)$ is the displacement of the *m*th cell in 2–10 cycles of saltatory movement in this averaging. This value is finite when the collective migration successfully conveys the cells in a specific direction. The methods of calculation of the set of the leading process direction $\{\boldsymbol{p}_m(t)\}$ and the set of the displacement $\{\boldsymbol{d}_m(t)\}$ through the simulation are given in the explanation of the mathematical model (see below).

As a model for calculating $P$ and $v$, we employ the cellular Potts model (Graner and Glazier, 1992; Graner, 1993; Glazier and Graner, 1993) consisting of the neurons, extracellular matrix (ECM), and obstacles. The probability of neuron configurations in this model depends on surface tension, volume-shape constriction, cell motility, cell–cell adhesion, and repulsion from obstacles. The cell configuration is expressed by the colored domains of Potts state on the lattice as shown in Fig. 2I,J. The Monte Carlo simulation of the cellular Potts model generates these states. The set of the leading process direction $\{\boldsymbol{p}_m(t)\}$, which corresponds to the direction of cell motion, is expressed by arrows in Fig. 2I,J. This direction obeys a conventional equation for a persistent random walk (Szabó et al, 2006; Kabla, 2012; Matsushita, 2017). The equation is integrated with the change in the set of the center of the cell $\{\boldsymbol{R}_m(t)\}$. From these centers, the displacement $\boldsymbol{d}_m(t)$ in Eq. (2) is

defined as $R_m(t) - R_m(t-1)$. The saltatory migration of neurons is expressed by the alternative changes in resting and migrating phases. The period of each phase is sampled from a normal distribution with a specific parameter set.

This mathematical model aims to examine the effects of cell–cell adhesion on collective migration to validate the experimental implication that the cell–cell adhesion of neurons affects their collective migration. Examining without other effects is desired in the simple examination without the possibility of other effects as far as possible. We use the virtual analysis in a mathematical model for such examination as follows.

We choose the cellular Potts model for this computer simulation (Graner and Glazier, 1992; Anderson et al, 2007; Hirashima et al, 2017). Because this model incorporates the cell–cell adhesion as an interface tension, this choice provides a tractable computer simulation method for this examination. We employ the model migration cells previously developed and consider their adhesion effect. The model cells are expressed below the energy of the cellular Potts model.

$$\mathcal{H}(\{m(\mathbf{r},t)\}, \{\mathbf{p}_m(\mathbf{r},t)\}, \{\mathbf{R}_m(\mathbf{r},t)\}) = \mathcal{H}_{st}(\{m(\mathbf{r},t)\})$$
$$+ \mathcal{H}_v(\{m(\mathbf{r},t)\}, \{\mathbf{p}_m(\mathbf{r},t)\})$$
$$+ \mathcal{H}_m(\{m(\mathbf{r},t)\}, \{\mathbf{p}_m(\mathbf{r},t)\}, \{\mathbf{R}_m(\mathbf{r},t)\})$$
$$+ \mathcal{H}_{cca}(\{m(\mathbf{r},t)\}, \{\mathbf{p}_m(\mathbf{r},t)\}, \{\mathbf{R}_m(\mathbf{r},t)\}) + \mathcal{H}_w(\{m(\mathbf{r},t)\}).$$
$$\text{(S1)}$$

Here, $\{m(\mathbf{r}, t)\}$ is the set of Potts state $m(\mathbf{r}, t)$ at the position $\mathbf{r}$ and time $t$. $m(\mathbf{r}, t)$ takes a number in 0, 1, 2, …, $N+1$ and indicates the index of the cell occupying $\mathbf{r}$ except for $m(\mathbf{r}, t) = 0$ and $N+1$. $m(\mathbf{r}, t) = 0$ expresses that the extracellular matrix occupies $\mathbf{r}$. $m(\mathbf{r}, t) = N+1$ expresses that there exists an obstacle at $\mathbf{r}$. $N$ is the number of cells. $N$ is maintained during the simulation and set to 48. We ignore the cell division or death in this simulation to avoid their additive effects on this evaluation of the cell–cell adhesion. $\{\mathbf{p}_m(\mathbf{r}, t)\}$ is the set of leading process direction, and a unit vector for the $m$th cell, $\mathbf{p}_m(\mathbf{r}, t)$, corresponds to the leading process direction of the neuron. $\{\mathbf{R}_m(t)\}$ is the set of cell centers, which is a slow degree of freedom equivalent to the center position of neurons in the static limit.

The positions $\mathbf{r}$ are defined at the lattice points on the two-dimensional square lattice with linear dimensions of $L_x$ in $x$-direction and $L_y$ in $y$-direction. The directions of $x$ and $y$ are orthogonal. In this work, we employ $L_x = 35$ and $L_y = 192$. Around $x = 0$ and 35, we put an array of obstacles in the $y$-direction and inhibit cell escape from the system in the $x$-direction as shown in Fig. 2I,J. We employ periodic boundary conditions in the $y$-direction to observe the long-distance migration, the distance of which is sufficiently longer than the typical cell size of 8 in this simulation.

The first term on the right-hand side of Eq. (S1) expresses the surface tension of cells and extracellular matrix (ECM).

$$\mathcal{H}_{st}(\{m(\mathbf{r},t)\}) = \gamma_E \sum_{\mathbf{rr'}} \eta_{m(\mathbf{r},t)m(\mathbf{r'},t)} [\theta_{m(\mathbf{r},t)}\delta_{m(\mathbf{r'},t)0} + \theta_{m(\mathbf{r'},t)}\delta_{m(\mathbf{r},t)0}]$$
$$+ \gamma_C \sum_{\mathbf{rr'}} \eta_{m(\mathbf{r},t)m(\mathbf{r'},t)}\theta_{m(\mathbf{r},t)}\theta_{m(\mathbf{r'},t)}$$
$$\text{(S2)}$$

Here, $\gamma_E$ represents the tension of the interface between ECM and cells, and $\gamma_C$ represents the tension of the interface between cells. In

this simulation, we set $\gamma_E = 3.0$ and $\gamma_C = 8.0$. These values of interface tensions stabilize the ECM between cells because ECM invades a separation between cells when $2\gamma_E < \gamma_C$. Therefore, these values enable us to simulate cell–cell contacts due to their additional adhesion. $\eta_{mn} = 1 - \delta_{mn}$ is unity for $m \neq n$, and, otherwise 0. This expresses the boundaries of Potts state domain, which express cell–cell or cell-ECM contact regions. $\theta_m = 1 - \delta_{m0} - \delta_{mN+1}$ is the indicator of cells, which distinguishes the cells from ECM and obstacles. $\delta_{mn}$ is Kronecker's delta. Namely, $\delta_{mn} = 1$ when $m = n$ and $\delta_{mn} = 0$ otherwise. The summation over site pair $\mathbf{r}$ and $\mathbf{r'}$ are taken for the neighboring pairs, which are conventionally defined by the pairs consisting of the nearest and next nearest sites (Graner and Glazier, 1992).

The second term on the right-hand side of Eq. (S1) expresses the volume and elongation shape constrictions.

$$\mathcal{H}_v(\{m(\mathbf{r},t)\}, \{\mathbf{p}_m(\mathbf{r},t)\}) = \kappa_0 V_0 \sum_m \left[1 - \frac{\sum_\mathbf{r} \delta_{mm(\mathbf{r},t)}}{V_0}\right]^2$$
$$+ \kappa_2 \sum_m \sum_\mathbf{r} [1 - \xi_{m(\mathbf{r},t)}(\mathbf{r})^2]\delta_{mm(\mathbf{r})}$$
$$\text{(S3)}$$

These terms keep volumes of cells around $V_0$ with the volume stiffness $\kappa_0$ and the elongation shape of cells in the direction of $\mathbf{p}_m$ with the strength of $\kappa_2$. We set both $V_0$ and $\kappa_0$ at 64.0; and $\kappa_2 = 5.0$. Here, $\xi_m(\mathbf{r}, t) = \mathbf{e}_m(\mathbf{r}) \cdot \mathbf{p}_m(t)$, is the direction cosine of $\mathbf{r}$ from the center of cell $\mathbf{R}_m(t)$ to $\mathbf{p}_m(t)$ for peripheral site $\mathbf{r}$ on the $m$th cell. This value is used for the strength of peripheral energy depending on $\mathbf{p}_m(t)$. $\mathbf{e}_m(\mathbf{r}, t)$ is the unit vector which is equal to $(\mathbf{r} - \mathbf{R}_m(\mathbf{r}, t))$ / $|\mathbf{r} - \mathbf{R}_m(\mathbf{r}, t)|$, where $\mathbf{R}_m(t)$ is the center of the $m$th cell at time $t$. The second term of the right-hand side is regarded as an elongation force version of anisotropic adhesion (Matsushita, 2017; Zajac et al, 2003; Vroomans et al, 2015; Matsushita, 2018). The formulation differs from the shape-rigidity in the anisotropic inertial term (Merks et al, 2006) in the sense of an anisotropic normal force on the cell membrane due to cytoskeleton elongation.

The third term on the right-hand side of Eq. (S1) expresses the motility of cells (Kabla, 2012; Matsushita, 2020).

$$\mathcal{H}_m(\{m(\mathbf{r},t)\}, \{\mathbf{p}_m(\mathbf{r},t)\}, \{\mathbf{R}_m(\mathbf{r},t)\})$$
$$= -\gamma_P \sum_{\mathbf{rr'}} [\xi_{m(\mathbf{r},t)}(\mathbf{r}) + \xi_{m(\mathbf{r'},t)}(\mathbf{r'})] \times \eta_{m(\mathbf{r},t)m(\mathbf{r'},t)}\theta_{m(\mathbf{r},t)}\theta_{m(\mathbf{r'},t)}.$$
$$\text{(S4)}$$

The strength of motility $\gamma_P$ is set to unity, which is less than values of tensions $\gamma_E$ and $\gamma_C$. We choose this value to inhibit interference of too-strong motility on interface tensions between cells. This term reproduces a cell migration only when the migrating cell contacts other cells at the neighboring site pair of $\mathbf{r}$ and $\mathbf{r'}$.

The fourth term on the right-hand side of Eq. (S1) expresses cell–cell adhesion.

$$\mathcal{H}_{cca}(\{m(\mathbf{r},t)\}, \{\mathbf{p}_m(\mathbf{r},t)\}, \{\mathbf{R}_m(\mathbf{r},t)\})$$
$$= -\Gamma_R \sum_{\mathbf{rr'}} \left[1 + \lambda\xi_{m(\mathbf{r},t)}(\mathbf{r})\right]\left[1 + \lambda\xi_{m(\mathbf{r'},t)}(\mathbf{r'})\right]$$
$$\times \eta_{m(\mathbf{r},t)m(\mathbf{r'},t)}\theta_{m(\mathbf{r},t)}\theta_{m(\mathbf{r'})}.$$
$$\text{(S5)}$$

$\delta$ is the strength of the cell–cell adhesion, and $\lambda$ is the imbalance parameter of the adhesion between the leading and following edges. We set $\lambda = 0.3$ with the simple assumption that the size of the cell

**The paper explained**

**Problem**

Regenerative medicine is expected to be a fundamental treatment for brain injury. However, conventional methods are invasive, and the therapeutic effect is limited. For many years, this has remained a major challenge to the clinical application of regenerative medicine.

**Results**

In this study, we show that the migration of new neurons produced from endogenous stem cells to the injured area is suppressed by neuraminidase, which is upregulated in the injured brain. Increased neuraminidase expression result in a decreased polysialic acid (PSA) level, excessive cell–cell adhesion, and reduced migration efficiency. Inhibition of neuraminidase by a clinically used neuraminidase inhibitor sustains PSA levels in new neurons and allows them to migrate to injured areas promoting neuronal regeneration and recovery of the brain function.

**Impact**

We demonstrated that a minimally invasive and broadly effective treatment of brain injury can be achieved by drug administration and proposed a novel treatment method for brain injury. Furthermore, we demonstrated the potential of drug repositioning, in which drugs used to treat influenza are applied to the treatment of brain injury.

body is larger than that of the growth cone. We control $\Gamma R$ for this examination and make data for Fig. 2L.

The fifth term on the right-hand side of Eq. (S1) expresses obstacles in this system.

$$\mathcal{H}_w(\{\boldsymbol{m}(\boldsymbol{r},t)\}) = \gamma w \sum_{\boldsymbol{rr}'} \eta_{m(\boldsymbol{r},t)m(\boldsymbol{r}',t)}$$
$$[\theta_{m(\boldsymbol{r},t)} \delta_{m(\boldsymbol{r}',t)N+1} + \theta_{m(\boldsymbol{r}',t)} \delta_{m(\boldsymbol{r},t)N+1}]. \quad (S6)$$

To reduce the cell motion through obstacles, we set $\gamma_W \gg \gamma_E$ and set $\gamma_W$ at 13.0 in this simulation. The cells have the dynamical degree of freedom, cell shape denoted by the domain of the Potts state $m(\boldsymbol{r}, t)$ and leading process direction $\boldsymbol{p}_m(\boldsymbol{r}, t)$ and $\boldsymbol{R}_m(\boldsymbol{r})$. In this simulation, these are updated by the Monte Carlo simulation with a Metropolis-Hastings algorithm (Hastings, 1970; Landau and Binder, 2005) as follows. The simulation calculates the consecutive time series of the cell configuration during $T$ Monte Carlo steps, which correspond to the dynamics of cells. We set the Monte Carlo step at 1/100 of the typical period of saltatory migration.

Saltatory migration is a well-known migration mode of neurons in their collective migration. This saltatory migration consists of two phases: one phase is the resting phase, in which the cell body stays immobile with the elongation of the leading process. The other phase is the migratory phase, in which the cell moves in the direction of the leading process with the shortening of the process. To mimic this motion, we assume the alternative changes between these phases of model cells. We consider the periods of these phases from the normal distribution with the average period and standard deviation of 50 Monte Carlo steps for both phases. The phases are randomly chosen for each cell at first, and then the period of these phases is sampled. The phase changes after this period, and then its period is sampled again. This procedure of phase change continues up to the end of the simulation. This set of parameters avoids cell jamming in a too-long resting phase. Furthermore, it avoids the inhibition of overtaking

between cells in the case of a too-long migrating phase. Overtaking between cells is observed in the experiment. Therefore, we choose this setting of the equivalent period distribution between the two phases.

The change of cell configuration from $\{m(\boldsymbol{r}, t)\}$ to $\{m(\boldsymbol{r}, t+1)\}$ is generated by one Monte Carlo step at time t conventionally consisting of $16L^2$ copies of Potts state from a site $\boldsymbol{r}$ to its neighboring site $\boldsymbol{r}'$ (Graner and Glazier, 1992). The copy is rejected in the resting phase. Otherwise, the site $\boldsymbol{r}$ is randomly selected from all sites, and $\boldsymbol{r}'$ is also randomly selected from the neighboring site of $\boldsymbol{r}$. The copy is accepted with the probability of Metropolis acceptance probability (Metropolis et al, 1953). For the state $s(t) = (\{m(\boldsymbol{r}, t)\}, \{\boldsymbol{p}_m(\boldsymbol{r}, t)\}, \{\boldsymbol{R}_m(\boldsymbol{r}, t)\})$, the probability $P$ is

$$P = \min[1, w(s(t+1)/w(s(t))], \quad (S7)$$

Here, $\{m(\boldsymbol{r}, t+1)\}, \{\boldsymbol{p}_m(\boldsymbol{r}, t+1)\}, \{\boldsymbol{R}_m(\boldsymbol{r}, t)\})$ is the state after the copy. $w(s(t))$, is the so-called Boltzman factor $\exp[-\beta H(\{m(\boldsymbol{r}, t)\}, \{\boldsymbol{p}_m(\boldsymbol{r}, t)\}, \{\boldsymbol{R}_m(\boldsymbol{r}, t)\})]$ with a control parameter of cell shape change, $\beta$. We set $\beta = 0.2$ to enable cells to deform their shape sufficiently for migration.

The change of the leading process direction $\boldsymbol{p}_m(t)$ per unit time obeys (Szabó et al, 2006; Kabla, 2012; Matsushita, 2017)

$$\Delta\boldsymbol{p}_m(t=1) = \frac{1}{aT}\left[\boldsymbol{d}_m\left(t+\frac{1}{2}\right) - \left(\boldsymbol{p}_m(t)\cdot\boldsymbol{d}_m\left(t+\frac{1}{2}\right)\right)\boldsymbol{p}_m(t)\right]. \quad (S8)$$

Here, we choose the leap-frog type discretization (Allen and Tildesley, 1989). $\tau$ is the persistent time of migration corresponding to the maintenance time of the leading process of a neuron. We set $\tau$ to 5 times the typical period of saltatory migration. $a$ is the discretization unit of space, and simply here, we set $a$ at unity. After the Monte Carlo step at time $t$, we calculate $\boldsymbol{d}_m(t+1/2)$ by $\Sigma_{\boldsymbol{r}}\boldsymbol{r}\delta_{mm(\boldsymbol{r},\,t+1)}/\Sigma_{\boldsymbol{r}}\delta_{mm(\boldsymbol{r},\,t+1)} - \boldsymbol{R}_m(t)$ and determine $\Delta\boldsymbol{p}_m$ through Eq. (S8). Then, we calculate $\boldsymbol{p}(t+1) = \hat{U}(\Delta\boldsymbol{p}_m) \boldsymbol{p}_m(t)$ through the rotation expressed by the orthogonal matrix $\hat{U}(\Delta\boldsymbol{p}_m)$ (Visscher and Feng, 2002; Matsushita et al, 2009). Here, $\hat{U}(\Delta\boldsymbol{p}_m)$ is evaluated using the quaternion method keeping its unity norm. After this procedure, we set $\boldsymbol{R}_m(t+1) = \Sigma_{\boldsymbol{r}}\boldsymbol{r}\delta_{mm(\boldsymbol{r},\,t+1)}/\Sigma_{\boldsymbol{r}}\delta_{mm(\boldsymbol{r},\,t+1)}$.

We generate the time series of cell configuration from this Monte Carlo procedure. From the cell configuration, we make movies and calculate the ordering indicator of P in Eq. (1), and average velocity $v$ in Eq. (2).

### For more information

Sawamoto lab website: https://k-sawamoto.com/.

## Data availability

This study includes no data deposited in external repositories.

The source data of this paper are collected in the following database record: biostudies:S-SCDT-10_1038-S44321-024-00073-7.

## Peer review information

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

## Acknowledgements

We thank Maiko Tanaka, Hiroshi Takase, Atsuko Imai, Mikio Furuse, Ai Kato, Kiyoshi Naruse, Chihong Song, Kazuyoshi Murata, Satoshi Ikeda, Itaru Imayoshi, Yukio Wakimoto, Hisashi Oishi and Takatoshi Ueki for technical support or comments, and Sawamoto laboratory members for discussions. We thank Bronwen Gardner from Edanz and Elizabeth Nakajima for editing a draft of this manuscript. This work was supported by research grants from Japan Agency for Medical Research and Development (AMED) (23gm1210007, 21bm0704033h0003 [to KS], 22gm6410007h0003, 23gm6410007h0003) [to CS]), Japan Society for the Promotion of Science (JSPS) (KAKENHI 20H05700, JP22H04926 [to K.S.]; 22K15202 [to MM], Bilateral Open Partnership Joint Research Projects, and Core-to-core Program "Neurogenesis Research & Innovation Center"), Cooperative Study Programs of National Institute for Physiological Sciences to KS, Takeda Science Foundation [to KS], Nitto Foundation [to MM] the Valencian Council for Innovation, Universities Science and Digital Society (PROMETEO/2019/075 [to JMGV]), and the Spanish Ministry of Science, Innovation and Universities (PCI2018-093062 [to VHP]). This study was supported by the Laboratory Animal Facility, Graduate

School of Medical Sciences, Nagoya City University, Research Equipment Sharing Center at Nagoya City University, and the Section of Electron Microscopy in the National Institute for Physiological Sciences.

## Author contributions

**Mami Matsumoto**: Formal analysis; Funding acquisition; Investigation; Writing—original draft. **Katsuyoshi Matsushita**: Formal analysis; Investigation; Methodology; Writing—original draft. **Masaya Hane**: Formal analysis; Investigation; Methodology; Writing—original draft. **Chentao Wen**: Software; Methodology. **Chihiro Kurematsu**: Investigation. **Haruko Ota**: Investigation; Methodology. **Huy Bang Nguyen**: Methodology. **Truc Quynh Thai**: Methodology. **Vicente Herranz-Perez**: Formal analysis; Visualization. **Masato Sawada**: Formal analysis; Supervision. **Koichi Fujimoto**: Supervision. **Jose Manuel Garcia-Verdugo**: Formal analysis; Supervision; Funding acquisition; Visualization; Methodology. **Koutarou D Kimura**: Supervision; Methodology. **Tatsunori Seki**: Formal analysis; Supervision; Investigation; Visualization; Writing—original draft. **Chihiro Sato**: Formal analysis; Supervision; Funding acquisition; Methodology; Writing—original draft. **Nobuhiko Ohno**: Supervision; Visualization; Methodology; Writing—original draft. **Kazunobu Sawamoto**: Conceptualization; Formal analysis; Supervision; Funding acquisition; Project administration; Writing—review and editing.

Source data underlying figure panels in this paper may have individual authorship assigned. Where available, figure panel/source data authorship is listed in the following database record: biostudies:S-SCDT-10_1038-S44321-024-00073-7.

## Disclosure and competing interests statement

The authors declare no competing interests.

# Expanded View Figures

**Figure EV1.  Semi-automatic segmentation of SBF-SEM images.**

(**A**) Overview of the semi-automatic segmentation flow incorporating deep learning-based cell area predictions and Seg2Link for segmentation. (**B–B″**) The automatic segmentation results of each new neuron (**B**), the updated segmentation results with additional cell boundaries (1 pixel in x-y plane) surrounding each neuron, which is required for accurately evaluating the non-adherent area and cell adhesion (**B′**), and the final segmentation results with manual corrections (**B″**). All the three steps were performed in Seg2Link. (**C–E′**) Representative electron microscopy images of new neurons in normal RMS by transmission electron microscopy (TEM) (**C**, **C′**), TEM of a high-pressure freeze-treated sample (High-pressure frozen) (**D**, **D'**), and serial block face scanning electron microscopy (SBF-SEM) (**E**, **E′**). Blue arrows indicate non-adherent areas. Scale bars: 2 μm (**C**, **D**, **E**), 500 nm (**C′**, **D′**, **E′**). (**F**) The distance in the non-adherent area between neighboring cells in normal RMS quantified using TEM ($n = 46$ cells), high-pressure freezing ($n = 51$ cells), and SBF-SEM ($n = 147$ cells); Mann–Whitney U test. (**G**) Percentage of adhesion or adherens junction (AJ)-like adhesions and non-adherent areas in normal RMS quantified using TEM, high-pressure freezing, and SBF-SEM (closed) and (open); each group $n = 5$ cells; Tukey multiple comparisons of means. (**H–I′**) Representative images of three-dimensional reconstruction of neuronal chains in adult RMS. H' and I' show adhesion area (red) between new neuron (yellow) and new neuron (green). The new neuron has a smooth morphology, with numerous small adhesions. In contrast, the irregularly shaped new neurons show large adhesions. The boxes in (**H**, **I**) are magnified in (**H′**, **I′**), respectively. Scale bars: 5 μm (**H**, **I**), 1 μm (**H′**, **I′**). Data information: In (**F**, **G**), data are presented as mean ± SEM. ***$P < 0.005$, ****$P < 0.001$.

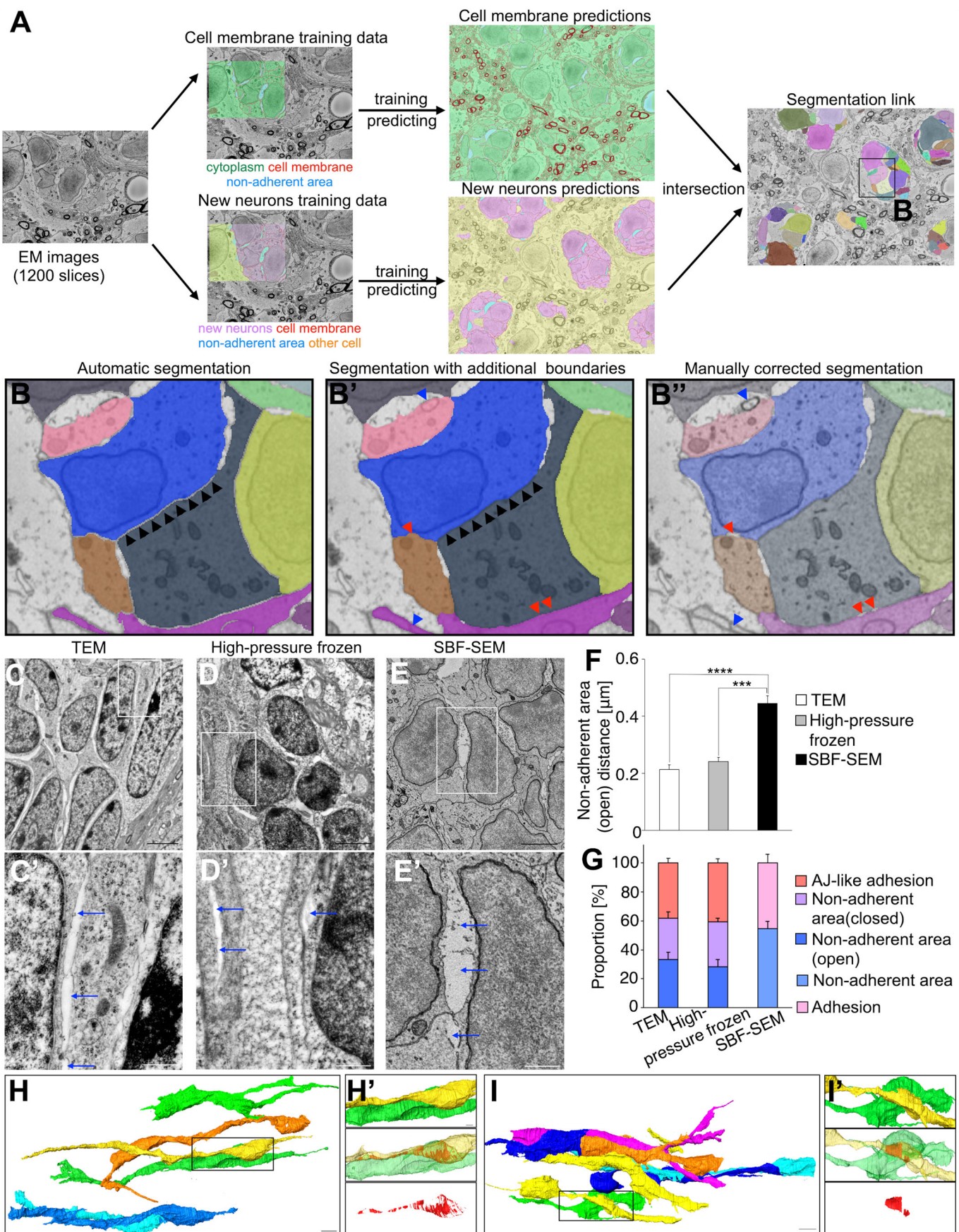

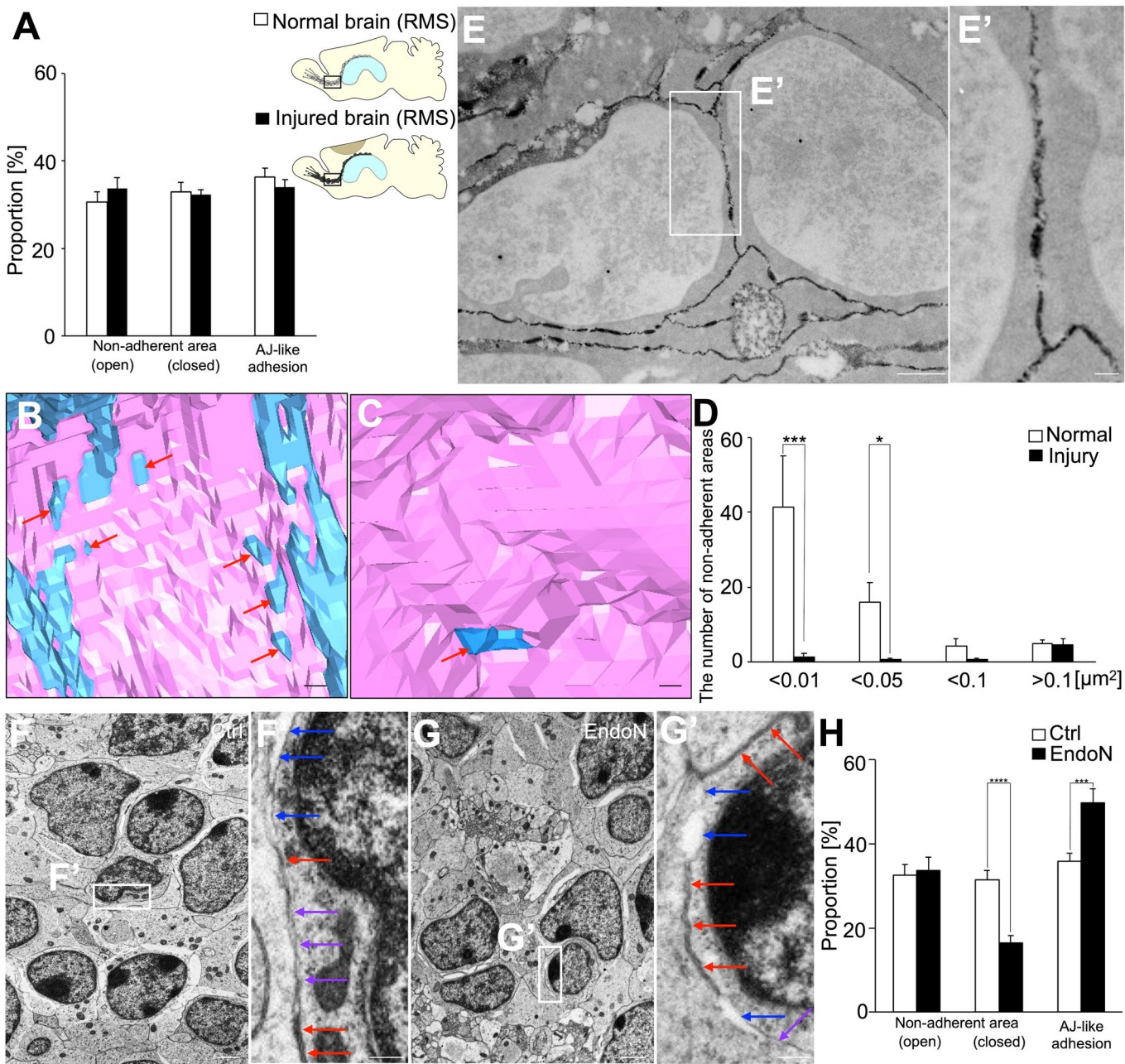

**Figure EV2. PSA regulates AJ-like adhesions and non-adherent areas in migrating new neurons.**

(A) Percentage of AJ-like adhesion and non-adherent areas ([open] or [closed]) in RMS in the normal brain and injured brain; each group $n = 30$ cells; non-adherent areas [open] and AJ-like adhesion; t-test, non-adherent areas [close]; Wilcoxon rank sum test. (B, C) Distribution of non-adhesion areas and adhesions to new neurons in neuronal cell bodies of adult RMS and injured brains. Pink and blue indicate adhesion to new neurons and non-adherent areas, respectively. Red arrows indicate pores. (B) and (C) show enlarged images of Fig. 2F' and G', respectively. Scale bars: 10 μm. (D) The number of non-adherent areas of each size in normal RMS ($n = 6$ adhesions) and injured brain ($n = 9$ adhesions); <0.01, <0.05, <0.1; Wilcoxon rank sum test, >0.1; t-test. (E, E') PSA-NCAM immunoelectron microscopy images of new neurons in RMS. The box in (E) is magnified in (E'). Scale bars: 1 μm (E), 200 nm (E'). (F–G') Representative transmission electron microscopy images of control (F, F') and endoN-treated (G, G') new neurons in RMS. Red, purple, and blue arrows indicate AJ-like adhesions, non-adherent areas (closed), and non-adherent areas (open), respectively; the boxes in (F, G) are magnified in (F', G'), respectively. Scale bars: 1 μm (E, F), 200 nm (F', G'). (H) Percentage of AJ-like adhesion and non-adherent areas ([open] or [closed]) in the control and endoN-treated groups. Removal of PSA from new neurons decreases the non-adherent area (closed) of new neurons and increases AJ-like adhesion with new neurons; each group $n = 30$ cells; non-adherent areas [open] and [close]; t-test, AJ-like adhesion; Wilcoxon rank sum test. Data information: In (A, D, H), data are presented as mean ± SEM. *$P < 0.05$, ***$P < 0.005$, ****$P < 0.001$; adjusted with Bonferroni correction in (A) and (H).

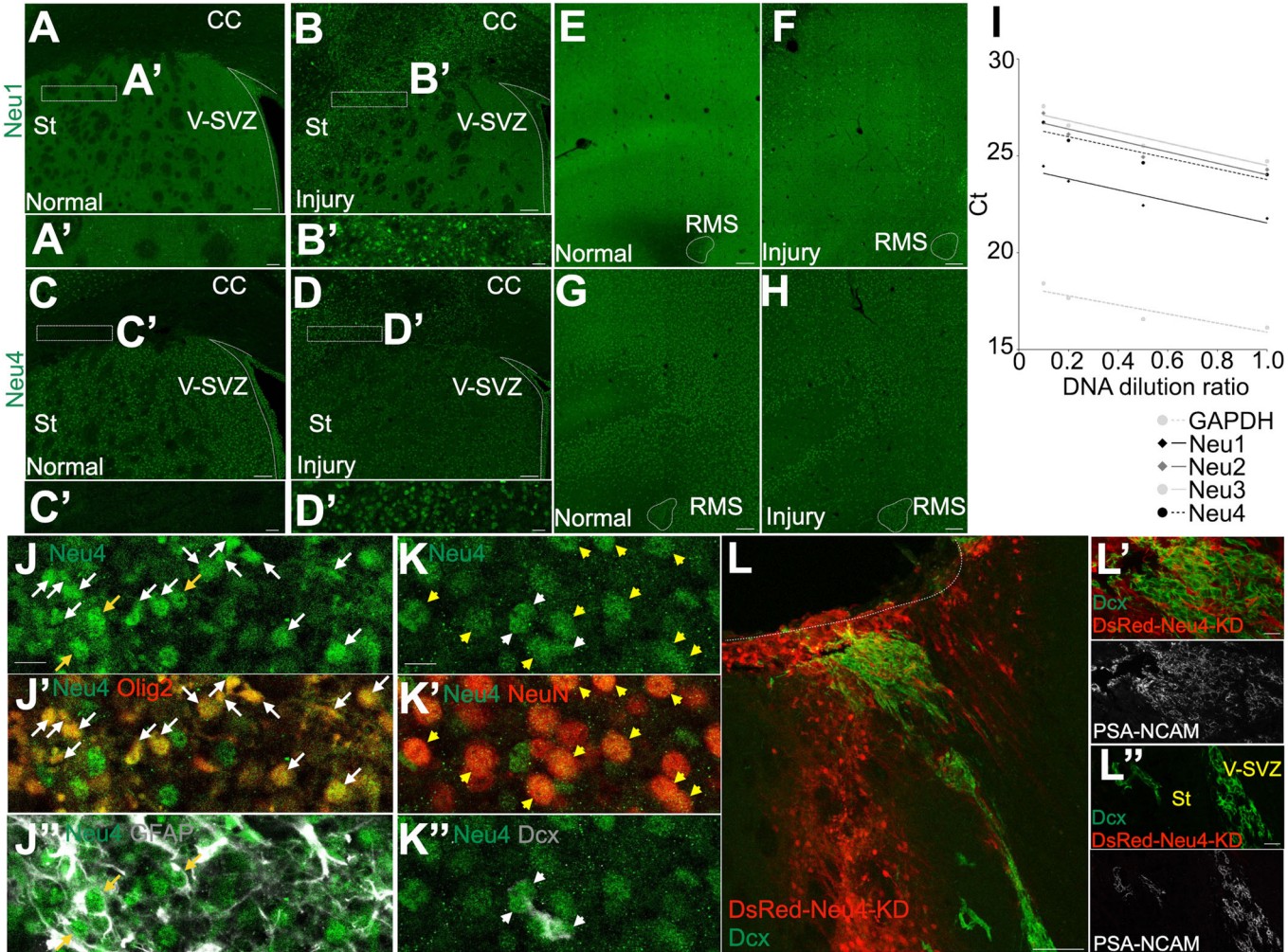

**Figure EV3. Neuraminidase expression is increased after brain injury.**

(A–B′) Representative images of brain sections stained for Neu1 (green) from normal (A) and injured (B) brain of WT mice. The boxed areas in (A and B) are enlarged in (A′ and B′), respectively. Scale bars: 100 μm (A, B), 20 μm (A′, B′). (C–D′) Representative images of brain sections stained for Neu4 (green) from normal (C) and injured (D) brain of WT mice. The boxed areas in (C and D) are enlarged in (C′ and D′), respectively. Scale bars: 100 μm (C, D), 20 μm (C′, D′). (E, F) Representative images of RMS sections stained for Neu1 (green) from normal (E) and injured (F) brain of WT mice. Scale bars: 100 μm (E, F). (G, H) Representative images of RMS sections stained for Neu4 (green) from normal (G) and injured (H) brain sections of WT mice. Scale bars: 100 μm (G, H). (I) Evaluation of RT-qPCR primer efficiencies. (J–J″) Representative images of injured brain sections of WT mice stained for Neu4 (green), Olig2 (red), and GFAP (white). Merged images of Neu4 and Olig2 in (J′) or Neu4 and GFAP in (J″). White and yellow arrows indicate Neu4+Olig2+ and Neu4+GFAP+ cells, respectively. Scale bars: 10 μm (J). (K–K″) Representative images of injured brain sections of WT mice stained for Neu4 (green), NeuN (red), and Dcx (white). Merged images of Neu4 and NeuN in (K′) or Neu4 and Dcx in (K″). Yellow and white arrows indicate Neu4+NeuN+ and Neu4+Dcx+ cells, respectively. Scale bars: 10 μm (K). (L–L″) Representative images of brain sections of WT mice injected with Neu4 KD virus 7 dpi and stained for Dcx (green) and DsRed (red) (L), or for Dcx (green), DsRed (red) and PSA-NCAM (white) (L′, L″) 21 dpi. Dotted lines indicate cortical lesion sites. Scale bars: 100 μm (L), 20 μm (L′, L″).

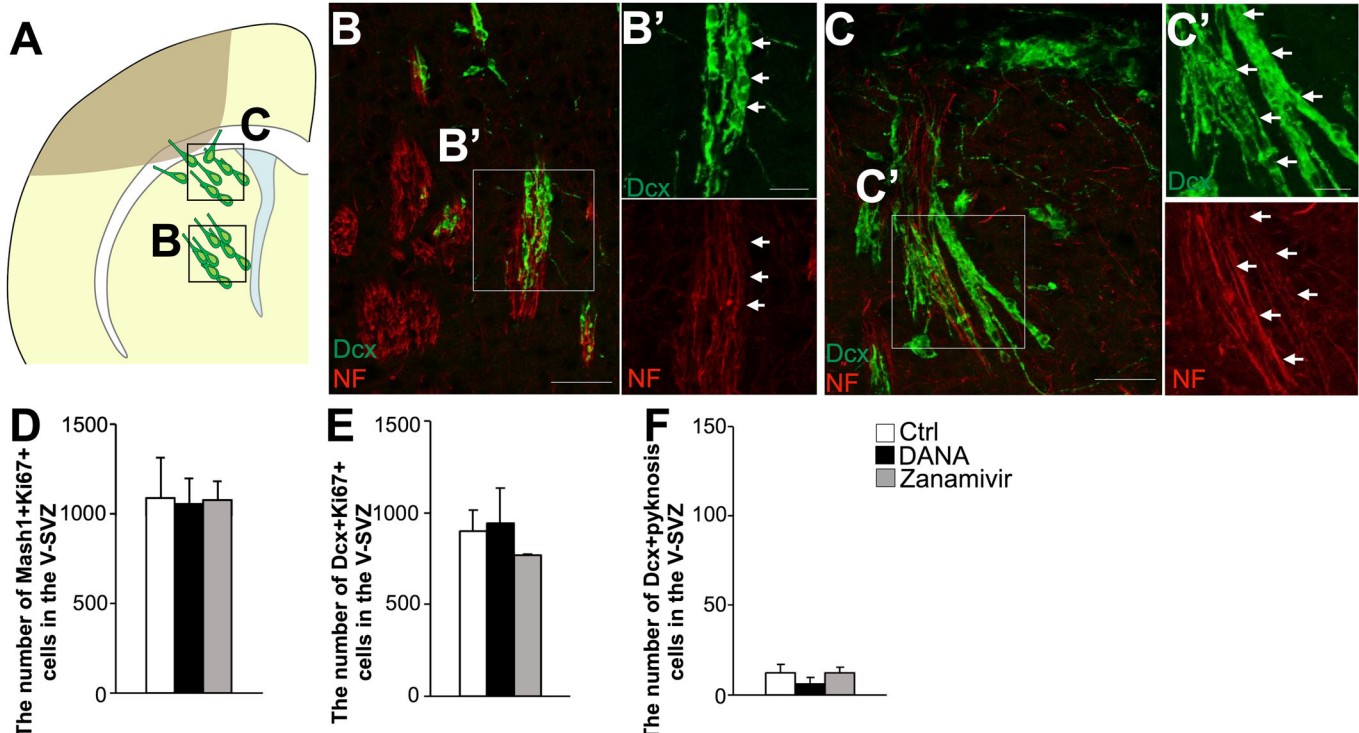

**Figure EV4. Neuraminidase inhibitors did not affect proliferation or cell death.**

(A) Experimental scheme indicating the brain areas (B, C). (B–C') Representative images of the middle part of the striatum (B) and the striatal tissue close to the injured site (C) stained for Dcx and NF from DANA-administered WT mice. The boxed areas in (B) and (C) are enlarged in (B') and (C'), respectively. Some new neurons were along the striatal fibers (arrows). Scale bars: 50 μm (B, C), 20 μm (B', C'). (D) The number of Mash1 + Ki67+ cells in the V-SVZ in control, DANA, and zanamivir groups; each group $n = 3$ mice; Tukey multiple comparisons of means. (E) The number of Dcx+Ki67+ cells in the V-SVZ in control, DANA, and zanamivir groups; each group $n = 3$ mice; Tukey multiple comparisons of means. (F) The number of Dcx+ pyknosis cells in the V-SVZ in control, DANA, and zanamivir groups; each group $n = 3$ mice; Tukey multiple comparisons of means. DANA or zanamivir administration in the injured brain did not affect proliferation or cell death. Data information: In (D, E, F), data are presented as mean ± SEM.

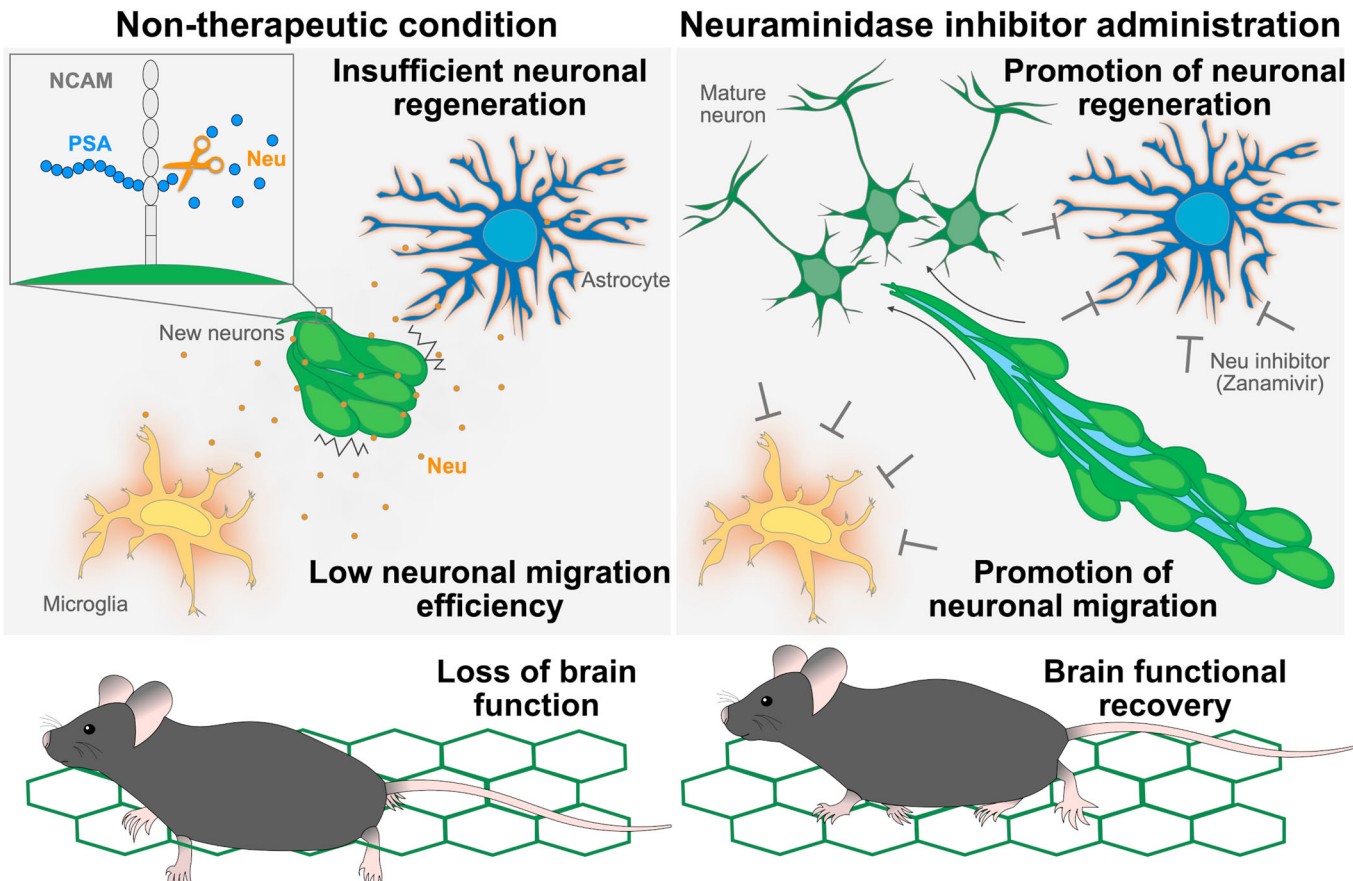

**Figure EV5.   Neuraminidase inhibitors promote neuronal migration, neuronal regeneration, and brain functional recovery.**

In non-therapeutic condition, activated glial cells release neuraminidase after brain injury, and released neuraminidase cleaves PSA of new neurons migrating toward the injured site, reduced PSA causes increased cell adhesion and decreased neuronal migration, and brain function is not recovered. Under condition of neuraminidase inhibitor administration, neuraminidase inhibitor promotes neuronal migration and regeneration, and brain function is recovered.

