## [Peer Review File · EMBO Molecular Medicine]

Neuraminidase inhibition promotes the collective migration of neurons and recovery of brain function

Mami Matsumoto, Katsuyoshi Matsushita, Masaya Hane, Chentao Wen, Chihiro Kurematsu, Haruko Ota, Huy Bang Nguyen, Truc Quynh Thai, Vicente Herranz-Perez, Masato Sawada, Koichi Fujimoto, Jose Garcia-Verdugo, Koutarou Kimura, Tatsunori Seki, Chihiro Sato, Nobuhiko Ohno, and Kazunobu Sawamoto

Corresponding author: Kazunobu Sawamoto (sawamoto@med.nagoya-cu.ac.jp)

Review Timeline:

Submission Date:	14th Dec 23
Editorial Decision:	17th Jan 24
Revision Received:	21st Mar 24
Editorial Decision:	5th Apr 24
Revision Received:	18th Apr 24
Accepted:	19th Apr 24

Editor: Zeljko Durdevic

Transaction Report:

17th Jan 2024

Dear Dr. Sawamoto,

Thank you for the submission of your manuscript to EMBO Molecular Medicine. We have now received feedback from the three reviewers who agreed to evaluate your manuscript. All three referees support publication of the manuscript and raise important but minor criticism that should be addressed in a revision of the current manuscript. If you would like to discuss further the points raised by the referees, I am available to do so via email or video. Let me know if you are interested in this option.

EMBO Molecular Medicine encourages a single round of revision only and therefore, acceptance or rejection of the manuscript will depend on the completeness of your responses included in the next, final version of the manuscript. For this reason, and to save you from any frustrations in the end, I would strongly advise against returning an incomplete revision. Also, please format the manuscript according to our "Author Guidelines": <https://www.embopress.org/page/journal/17574684/authorguide>

We would welcome the submission of a revised version within three months for further consideration. Please let us know if you require longer to complete the revision.

I look forward to receiving your revised manuscript.

Yours sincerely,

Zeljko Durdevic

*** Instructions to submit your revised manuscript ***

- 1) a .docx formatted version of the manuscript text (including Figure legends and tables)
- 2) Separate figure files*
- 3) supplemental information as Expanded View and/or Appendix. Please carefully check the authors guidelines for formatting Expanded view and Appendix figures and tables at <https://www.embopress.org/page/journal/17574684/authorguide#expandedview>
- 4) a letter INCLUDING the reviewer's reports and your detailed responses to their comments (as Word file).
- 5) The paper explained: EMBO Molecular Medicine articles are accompanied by a summary of the articles to emphasize the major findings in the paper and their medical implications for the non-specialist reader. Please provide a draft summary of your

article highlighting

6) For more information: There is space at the end of each article to list relevant web links for further consultation by our readers. Could you identify some relevant ones and provide such information as well? Some examples are patient associations, relevant databases, OMIM/proteins/genes links, author's websites, etc...

7) Author contributions: the contribution of every author must be detailed in a separate section.

8) EMBO Molecular Medicine now requires a complete author checklist (<https://www.embopress.org/page/journal/17574684/authorguide>) to be submitted with all revised manuscripts. Please use the checklist as guideline for the sort of information we need WITHIN the manuscript. The checklist should only be filled with page numbers where the information can be found. This is particularly important for animal reporting, antibody dilutions (missing) and exact values and n that should be indicated instead of a range.

9) Every published paper now includes a 'Synopsis' to further enhance discoverability. Synopses are displayed on the journal webpage and are freely accessible to all readers. They include a short stand first (maximum of 300 characters, including space) as well as 2-5 one sentence bullet points that summarise the paper. Please write the bullet points to summarise the key NEW findings. They should be designed to be complementary to the abstract - i.e. not repeat the same text. We encourage inclusion of key acronyms and quantitative information (maximum of 30 words / bullet point). Please use the passive voice. Please attach these in a separate file or send them by email, we will incorporate them accordingly.

You are also welcome to suggest a striking image or visual abstract to illustrate your article. If you do please provide a jpeg file 550 px-wide x 300-800px high.

10) A Conflict of Interest statement should be provided in the main text

11) Please note that we now mandate that all corresponding authors list an ORCID digital identifier. This takes <90 seconds to complete. We encourage all authors to supply an ORCID identifier, which will be linked to their name for unambiguous name identification.

Currently, our records indicate that the ORCID for your account is 0000-0003-1984-5129.

Please click the link below to modify this ORCID:
Link Not Available

Graphs 800-1,200 DPI
Photos 400-800 DPI
Colour (only CMYK) 300-400 DPI"

*Additional important information regarding figures and illustrations can be found at <https://bit.ly/EMBOPressFigurePreparationGuideline>. See also figure legend preparation guidelines: <https://www.embopress.org/page/journal/17574684/authorguide#figureformat>

***** Reviewer's comments *****

Referee #1 (Comments on Novelty/Model System for Author):

I suggest you ask a mathematical modeler to review that part of the paper. Although it seems logical I am not an expert on this aspect of the paper.

Referee #1 (Remarks for Author):

Review of: Neuraminidase inhibition promotes the collective migration of neurons and recovery of brain function

This manuscript explores the functional role of PSA-NCAM in functional repair of brain injury. The role of reduced PSA-NCAM expression in chain migration in trauma is interesting and may help explain the relatively low level of repair and is thus interesting and important to consider. They use Dcx-GFP mice as well as nestinCreERT2::NSE diphtheria toxin mice in order to remove V-SVZ cells and to show they are necessary for functional improvement. They also use common marmosets to show these mechanisms are at play in primates. They use a well-thought through semi-automatic imaging and segmentation approach to quantify adhesion. They use serial sectioning and EM and then render three dimensional images that can be quantified. They then use deep learning, mathematical modelling and quantification of directed versus entropic migration.

Using these powerful approaches, the authors show that there are more non-adherent areas between newborn neurons than between neurons and astrocytes. They show that even in adherent areas you can find small "pores" of non-adherence. They also show that brain injury increases the proportion of adherent areas and decreases pores between neuroblasts suggesting this may restrict repair. They also show that endogenous neuroaminidase can regulate adhesion and therefore migration of SVZ neuroblasts. Decreasing Neu1 with knockdown massively increased emigration to injury and PSA-NCAM expression. Also neuraminidase inhibitors increased migration to injury.

Overall, this is a fascinating set of results using novel and powerful approaches which may have significant effects on the field. The writing is very good and the images are compelling. I recommend publication after a few issues laid out below are tackled. The findings point to a novel druggable target to increase endogenous neuroblast migration to injury - neuraminidase inhibitors.

Major issues

1. The authors state in line 468 that: "suggesting that the non-adherent areas are not dependent on fixation methods (Figure S1C-F)." However the frames selected in C'-E' and quantification in S1G clearly show significant differences. Please re-write to make this clear.
2. The authors suggest that after injury the chains are composed of neuroblasts with more complex morphology (Fig. 2B).
 - a. Please specify if these are chains in the SVZ or in areas of ectopic migration or both. It is not clear from the annotation ex. "injured site <- V-SVZ.
 - b. Also, it would help if there was some quantification of this phenomenon.
 - c. Finally Nam et al quantified speed of neuroblast migration and correlated it with complexity of cell migration (PMID 17853439), this should be referenced.
 - d. The comparison between VideoS2 and S3 are beautiful and show these differences well, however they are qualitative impressions and would be strengthened by quantification. (also the final 30 seconds or so of S3 could be cropped out as they are stationary.
3. The finding that PSA-NCAM levels are decreased after injury is unexpected as usually injury is associated with increased expression of PSA-NCAM which is a marker of newborn neurons. Is there increased rates of neurogenesis in marmoset after injury? You show that the intensity of PSA-NCAM immunofluorescence was decreased in injury but was the number of PSA-NCAM+ cells altered?

Minor issues

4. What exactly is Fig. 2I showing? This is unclear to me.
5. The mathematical modelling is very cool but please change the title to "Mathematical modelling suggests efficient chain migration depends on appropriate cell adhesion levels".
6. Please explain Fig. 3 H pictorial representation of experiment. Is the orange band the "penumbra" of injury or is it where the virus was injected, or both? What does the narrow orange arrow point to?
7. Arrows in figures should touch the element they are pointing to. For example in Fig S2E'.
8. Some of the Dcx (green) immunofluorescence in the large panels in Fig. 4C and D look like they might be labelling striatal fibre tracts ("pencil fibres"). Please address this possibility.
9. Most of the figures appear to be very "busy" or "crowded" - please increase the amount of space between them.

Referee #2 (Comments on Novelty/Model System for Author):

The study developed a conceptually novel approach to improve brain function recovery upon brain injury. It is a broad study with a lot of data that support its conclusions. The selection of animal models is appropriate, especially inclusion of marmoset sets this study apart from traditional studies focusing on rodents.

Referee #2 (Remarks for Author):

The study by Matsumoto et al. focuses on the role of PSA modification of NCAM adhesion molecule during neuronal migration from the SVZ to a site of brain injury. The authors utilize a novel EM based method to quantify adhesion between neurons in the RMS revealing two types of spaces between adjacent migrating neurons - closed and opened non-adherent regions. They have developed a machine learning based image analysis method to quantify these two regions and determined that removal of PSA results in a decrease of closed non-adherent regions, pointing to the role of PSA in maintaining this structure. This extends a series of prior studies on the role of PSA in neuronal migration from SVZ to olfactory bulb. In the second part of the manuscript, the authors focus on the role of PSA in neuronal migration from the SVZ to a site of injury. They employ light induced brain injury model and demonstrate that the level of PSA on neurons migrating towards the injury site is decreased. They perform several genetic and pharmacological manipulations to demonstrate that this decrease is caused by an increase in the expression of endoneuraminidases and decrease in PSA level on migrating young neurons. Finally, they show that administration of two pharmacological inhibitors of neuraminidases to injured mice results in improved neuronal migration and functional recovery following brain injury. Similar increase in neuronal migration is also observed in an injured marmoset brain, suggesting that application of this treatment might be beneficial for patients with brain injury.

This is an interesting study with a direct potential clinical impact. However, I find that the authors do not introduce in sufficient detail previous studies examining the role of PSA in the mouse brain. Namely, they omitted to cite a study PMID: 20237270 that is highly relevant to their work. This paper deserves to be discussed, as the authors suggest that removal of PSA can result in slowing down of neuronal migration along the RMS with a concomitant increase in neuronal dispersion from the SVZ. Thus, removal of PSA could paradoxically also enhance recovery after brain injury, by decreasing neuronal migration from SVZ to olfactory bulb. Below are few addition points that should be addressed before publication.

1. How far from the injury site do you observe increase in neuraminidases expression? Does it reach to SVZ and RMS? Do you see any effects on non-adherent and adherent zones between migrating neurons in the RMS upon injury?
2. How exactly is PSA intensity measured upon the drug treatments? Is it intensity per cell or intensity per section? Increase of intensity per section could simply reflect increase in the number of migrating neurons. It is therefore important to quantify whether PSA intensity per individual migrating neurons is also increased.
3. Figure 3: "(I-J)Representative images of control (J-J")and Neu1 KD(K-K") virus injected" check panel callouts - these callouts are incorrect
4. Figure 6G - are the conditions switched? The graph suggests that zanamivir treatment decreases neuronal migration.

Referee #3 (Comments on Novelty/Model System for Author):

Model systems, particularly the behavioural studies was carried out in rodent, and non-primate models, which makes the findings very compelling

Referee #3 (Remarks for Author):

Dear Editor, Find attached my comments on the manuscript entitled "Neuraminidase inhibition promotes the collective migration of neurons and recovery of brain function" by Matsumoto et al.

Overall, I found this study well written and excellently carried out, and the novel findings exciting for future drug treatment for treating brain injuries. The study encapsulates the whole spectrum of techniques, from basic immunohistochemistry, EM, through to modelling, through to behavioural studies, both in rodent and non-primate models, which tantalisingly show some promise in disease repair. Which is the exciting finding with this paper. Thus I would wholly recommend publishing this paper your journal.

My only major criticism would be of the figures as a whole. And this is purely stylistic, as I'm not a great fan of having to read the figure legends to find out what I'm looking at. I much prefer to have the figures well annotated, so you can see what is going on, without having to refer to the legends

Particularly figures 1 & 2, without any annotation it is difficult to understand what is going on, in the very colourful figure!

Minor comment in figure 3, levels of Neu1 and neu4 show an increase following injury, but could you comment on why you

think the levels of neu2 significantly goes down?

Responses to the Reviewers

Referee #1 (Comments on Novelty/Model System for Author):

I suggest you ask a mathematical modeler to review that part of the paper. Although it seems logical I am not an expert on this aspect of the paper.

Referee #1 (Remarks for Author):

Review of: Neuraminidase inhibition promotes the collective migration of neurons and recovery of brain function

This manuscript explores the functional role of PSA-NCAM in functional repair of brain injury. The role of reduced PSA-NCAM expression in chain migration in trauma is interesting and may help explain the relatively low level of repair and is thus interesting and important to consider. They use Dcx-GFP mice as well as nestinCreERT2::NSE diphtheria toxin mice in order to remove V-SVZ cells and to show they are necessary for functional improvement. They also use common marmosets to show these mechanisms are at play in primates. They use a well-thought through semi-automatic imaging and segmentation approach to quantify adhesion. The use serial sectioning and EM and then render three dimensional images that can be quantified. They then use deep learning, mathematical modelling and quantification of directed versus entropic migration.

Using these powerful approaches, the authors show that there are more non-adherent areas between newborn neurons than between neurons and astrocytes. They show that even in adherent areas you can find small "pores" of non-adherence. They also show that brain injury increases the proportion of adherent areas and decreases pores between neuroblasts suggesting this may restrict repair. They also show that endogenous neuroaminidase can regulate adhesion and therefore migration of SVZ neuroblasts. Decreasing Neu1 with knockdown massively increased emigration to injury and PSA-NCAM expression. Also neuraminidase inhibitors increased migration to injury.

Overall, this is a fascinating set of results using novel and powerful approaches which may have significant effects on the field. The writing is very good and the images are

compelling. I recommend publication after a few issues laid out below are tackled. The findings point to a novel druggable target to increase endogenous neuroblast migration to injury - neuraminidase inhibitors.

We thank the reviewer for their deep insights and positive and constructive comments on this paper. We have made modifications to the manuscript, figure, reference, and video in response to the comments received. Detailed responses to specific comments are as follows:

Major issues

1. The authors state in line 468 that: "suggesting that the non-adherent areas are not dependent on fixation methods (Figure S1C-F)." However the frames selected in C'-E' and quantification in S1G clearly show significant differences. Please re-write to make this clear.

Thank you for your comments. We apologize for the lack of description of the results. In response to your comment, we have rewritten the description of the results as follows :

"The non-adherent areas were observed not only in transmission electron microscopy (TEM) and SBF-SEM images after chemical fixation, but also in samples prepared with high-pressure freezing followed by freeze-substitution (Vanhrreveld, A., Crowell, J., & Malhotra, 1965) (Fig EV1C-E'). The distance in the non-adherent area between neighboring cells in SBF-SEM was greater than that in TEM and high-pressure freezing, but the proportion of non-adherent area did not differ among all fixation methods (Fig EV1F, G). (Page 6, lines 145-152)"

2. The authors suggest that after injury the chains are composed of neuroblasts with more complex morphology (Fig. 2B).

a. Please specify if these are chains in the SVZ or in areas of ectopic migration or both. It is not clear from the annotation ex. "injured site <-- V-SVZ.

We apologize for the insufficient description. This is a neuronal chain migrating in the injured site. We added the following description to show that they are new neurons migrating in the ectopic areas, as you indicated.

“(A-B’’) Representative images of three-dimensional re-constructions of neuronal chain in the adult RMS (A) and injured site (B). (Page 34, lines 1069-1070)”

b. Also, it would help if there was some quantification of this phenomenon.

Thank you for the suggestion to quantify these data. Regarding this phenomenon, we quantified the orientation of new neurons in chains and the length of the leading process of new neurons. In the injured brain, the percentage of new neurons in the forward direction is decreased, whereas the percentages in the reverse and in other directions are increased. We also found that the process length of new neurons is also shorter in chains in the injured brain. Based on these results, we have added the following description and included the new data in Figure 2C–E.

“Chains in the normal brain showed an aligned cell orientation and new neurons exhibited smooth morphologies. However, chains in the injured brain had more random cell orientations and new neurons had a shorter leading process (Figure 2A–E, Movie EV2–3). (Page 7, lines 168-171)”

c. Finally Nam et al quantified speed of neuroblast migration and correlated it with complexity of cell migration (PMID 17853439), this should be referenced.

Thank you for your comments. Based on your comments, we have added the following statement and cited the suggested paper as follows.

“Considering a report showing no correlation between neuron morphology and migration speed(Nam et al, 2007), irregular orientation of new neurons does not necessarily indicate low migration speed. However, irregular orientation may prevent new neurons from moving efficiently and effectively to their destination, the site of injury. (Page 7, lines 171-175)”

d. The comparison between VideoS2 and S3 are beautiful and show these differences well, however they are qualitative impressions and would be strengthened by quantification. (also the final 30 seconds or so of S3 could be cropped out as they are stationary.

Thank you for reviewing these videos. As we responded to your comment “b”, we have quantified the difference in the orientation and leading process length of new neurons in chains between normal and injured brain and added the data in Fig. 2C–F. Thank you very much for your kind attention to our mistake about Video S3. As you suggested, we have cropped out the parts of the video where the image was stationary.

3. The finding that PSA-NCAM levels are decreased after injury is unexpected as usually injury is associated with increased expression of PSA-NCAM which is a marker of newborn neurons. Is there increased rates of neurogenesis in marmoset after injury? You show that the intensity of PSA-NCAM immunofluorescence was decreased in injury but was the number of PSA-NCAM+ cells altered?

We apologize for the insufficient description here. As you commented, we think that the expression of PSA-NCAM per brain tissue outside of the V-SVZ is increased by the migration of PSA-NCAM+ cells to the injured site after injury. On the other hand, our results indicate a decrease in the intensity of PSA-NCAM per cell, not per brain tissue. Following your comment, we counted the number of PSA-NCAM+ cells and found that they were increased in the ipsilateral hemisphere compared to contralateral hemisphere after brain injury in both mice and marmosets. We have added these new data in Figs. 2P and 6E, and revised the description as follows.

“Although the number of PSA-NCAM+ cells was increased by brain injury, PSA signal intensity in each new neuron was lower in the injured brain than in the normal brain (Figure 2M–P). (Page 8, lines 211-213)”

“The number of PSA-NCAM+ cells outside of the V-SVZ was increased by brain injury. However, the PSA signal intensity of these new neurons migrating to the brain injury was lower than that of new neurons migrating through the RMS, but was maintained in zanamivir-treated animals (Figure 6B–E). (Page 12, lines 354-358)”

Minor issues

4. What exactly is Fig. 2I showing? This is unclear to me.

We apologize for the insufficient explanation. We have added “order indicator”, “strength of adhesion” and “average velocity” in Fig. 2I (currently Fig. 2L)

5. The mathematical modelling is very cool but please change the title to "Mathematical modelling suggests efficient chain migration depends on appropriate cell adhesion levels".

Thank you for your constructive comments. We have modified the title as you suggested.

6. Please explain Fig. 3 H pictorial representation of experiment. Is the orange band the "penumbra" of injury or is it where the virus was injected, or both? What does the narrow orange arrow point to?

We apologize for the confusing figure in the original manuscript. The orange area represents the virus-infected area proximal to the injured site. The "orange arrow" was not an arrow, but a needle for virus injection. We have modified the schematic to show the position of the infected area in the brain more precisely.

7. Arrows in figures should touch the element they are pointing to. For example in Fig S2E'.

We apologize for the confusing arrows in the original figures. As you suggested, we have repositioned the arrows and arrowheads for all figures.

8. Some of the Dcx (green) immunofluorescence in the large panels in Fig. 4C and D look like they might be labelling striatal fibre tracts ("pencil fibres"). Please address this possibility.

Thank you very much for your very interesting comments. We double-stained the brain sections for Dcx and Neurofilament, which is expressed in striatal fiber tracts ("pencil fibers"), and found that the striatal fiber tracts were not labeled with the Dcx antibody. Interestingly, some of the Dcx+ cells were observed to be migrating along the pencil fibers to the injured site. We have added these new data in Figure EV4A–C'.

"(B–C') Representative images of the middle part of the striatum (B) and the striatal tissue close to the injured site (C) stained for Dcx and NF from DANA-administered WT

mice. The boxed areas in (B) and (C) are enlarged in (B') and (C'), respectively. Some new neurons were along the striatal fibers (arrows). (Page 44, lines 1332-1335)”

9. Most of the figures appear to be very "busy" or "crowded" – please increase the amount of space between them.

Thank you for your constructive comments regarding the arrangement of Figures. As you indicated, we have increased the amount of space between figures.

Referee #2 (Comments on Novelty/Model System for Author):

The study developed a conceptually novel approach to improve brain function recovery upon brain injury. It is a broad study with a lot of data that support its conclusions. The selection of animal models is appropriate, especially inclusion of marmoset sets this study apart from traditional studies focusing on rodents.

Referee #2 (Remarks for Author):

The study by Matsumoto et al. focuses on the role of PSA modification of NCAM adhesion molecule during neuronal migration from the SVZ to a site of brain injury. The authors utilize a novel EM based method to quantify adhesion between neurons in the RMS revealing two types of spaces between adjacent migrating neurons - closed and opened non-adherent regions. They have developed a machine learning based image analysis method to quantify these two regions and determined that removal of PSA results in a decrease of closed non-adherent regions, pointing to the role of PSA in maintaining this structure. This extends a series of prior studies on the role of PSA in neuronal migration from SVZ to olfactory bulb. In the second part of the manuscript, the authors focus on the role of PSA in neuronal migration from the SVZ to a site of injury. They employ light induced brain injury model and demonstrate that the level of PSA on neurons migrating towards the injury site is decreased. They perform several genetic and pharmacological manipulations to demonstrate that this decrease is caused by an increase in the expression of endoneuraminidases and decrease in PSA level on migrating young neurons. Finally, they show that administration of two pharmacological inhibitors of neuraminidases to injured mice results in improved neuronal migration and functional recovery

following brain injury. Similar increase in neuronal migration is also observed in an injured marmoset brain, suggesting that application of this treatment might be beneficial for patients with brain injury. This is an interesting study with a direct potential clinical impact. However, I find that the authors do not introduce in sufficient detail previous studies examining the role of PSA in the mouse brain. Namely, they omitted to cite a study PMID: 20237270 that is highly relevant to their work. This paper deserves to be discussed, as the authors suggest that removal of PSA can result in slowing down of neuronal migration along the RMS with a concomitant increase in neuronal dispersion from the SVZ. Thus, removal of PSA could paradoxically also enhance recovery after brain injury, by decreasing neuronal migration from SVZ to olfactory bulb. Below are few addition points that should be addressed before publication.

Thank you very much for your valuable comments. We have added the following statements to the discussion as you suggested.

“The idea of injecting a drug that removes PSA into the brain and using the dispersed neurons from the V-SVZ for treatment has also been reported(Battista & Rutishauser, 2010), but this method would be invasive and would causes loss of PSA even in normal neurons. (Page 15, lines 448-452)”

1. How far from the injury site do you observe increase in neuraminidases expression? Does it reach to SVZ and RMS? Do you see any effects on non-adherent and adherent zones between migrating neurons in the RMS upon injury?

Thank you for your comments. We have added a more wide-ranging image of the brain tissue to show how far away the increased neuraminidase expression reached. The increased neuraminidase expression caused by brain injury did not reach the V-SVZ or the RMS. Non-adherent areas and adherent zones of new neurons in the RMS in the injured brain were not different from those of new neurons migrating in the RMS of the normal brain. These new results were described as follows and included in Figures EV2A and EV3A-H.

“The increase in neuraminidase expression due to brain injury did not reach the V-SVZ or RMS (Figure EV3A–H). (Page 9, lines 249-250)”

“Brain injury decreases non-adherent area of new neurons at the injured site but not in the RMS (Figure 2F–F, EV2A). (Page 7, lines 175-177)”

2. How exactly is PSA intensity measured upon the drug treatments? Is it intensity per cell or intensity per section? Increase of intensity per section could simply reflect increase in the number of migrating neurons. It is therefore important to quantify whether PSA intensity per individual migrating neurons is also increased.

Thank you for your comment. We apologize for the unclear descriptions of the methods we used for the analysis. We quantified PSA intensity per individual migrating new neuron. We have modified the description as follows.

“PSA signal intensity in each new neuron was lower in the injured brain than in the normal brain (Figure 2M–P). (Page 8, lines 211-213)”

3. Figure 3: "(I-J) Representative images of control (J-J') and Neu1 KD(K-K') virus injected" check panel callouts - these callouts are incorrect

We apologize for those confusing images. Fig 3I, Fig 3J and Fig S3H needed to show all Dcx+ cells in the sections to show that KD of Neuraminidase increased the number of new neurons in the injured area. On the other hand, Fig 3I', Fig 3J' and Fig S3H' contained fewer z-images than Fig 3I, Fig J and Fig S3H, in order to show the increased PSA signal of new neurons in the injured site with Neuraminidase KD. This is because the PSA antibody only stains the surface of the sections. The original manuscript incorrectly stated that Fig 3I', Fig 3J' and Fig S3H' were enlarged images of Fig 3I, Fig 3J and Fig S3H, respectively. We apologize for the imprecise descriptions in their figure legends. To avoid misunderstanding, we have removed the PSA-NCAM signals from Fig 3I, Fig 3J and Fig S3H (Fig 3I, Fig 3J and Fig EV3L in the revised manuscript), and edited their Figure legends as follows.

“(I–J’’) Representative images of brain tissues injected with control (I) or Neu1-KD (J) virus at 7 days post injury (dpi) and stained for Dcx (green) and DsRed (red) at 21 dpi. Representative images of brain tissues injected with control (I’, I’’) or Neu1-KD (J’, J’’) virus at 7 dpi and stained for Dcx (green), DsRed (red), and PSA-NCAM (white) at 21 dpi. (Page 36, lines 1146-1150)”

“(L–L’’) Representative images of brain sections of brain tissues injected with Neu4 KD virus 7 dpi and stained for Dcx (green) and DsRed (red) (L), or for Dcx

(green), DsRed (red) and PSA-NCAM (white) (L', L'') 21 dpi. Dotted lines indicate cortical lesion sites. (Page 43, lines 1321-1324)''

4. Figure 6G - are the conditions switched? The graph suggests that zanamivir treatment decreases neuronal migration.

We apologize for the confusing description of the data. This data shows that zanamivir administration allowed the new neurons to migrate closer to the injured site. Therefore, we have changed the title of this graph to "Distance to injured site".

Referee #3 (Comments on Novelty/Model System for Author):

Model systems, particularly the behavioural studies was carried out in rodent, and non-primate models, which makes the findings very compelling

Referee #3 (Remarks for Author):

Dear Editor, Find attached my comments on the manuscript entitled "Neuraminidase inhibition promotes the collective migration of neurons and recovery of brain function" by Matsumoto et al. Overall, I found this study well written and excellently carried out, and the novel findings exciting for future drug treatment for treating brain injuries. The study encapsulates the whole spectrum of techniques, from basic immunohistochemistry, EM, through to modelling, through to behavioural studies, both in rodent and non-primate models, which tantalisingly show some promise in disease repair. Which is the exciting finding with this paper. Thus I would wholly recommend publishing this paper your journal.

My only major criticism would be of the figures as a whole. And this is purely stylistic, as I'm not a great fan of having to read the figure legends to find out what I'm looking at. I much prefer to have the figures well annotated, so you can see what is going on, without having to refer to the legends. Particularly figures 1 & 2, without any annotation it is difficult to understand what is going on, in the very colourful figure!

We apologize for the complicated figure structure. As you suggested, we have inserted schematic diagrams in Figs 1, 2, EV2 and EV4 to make it easier to understand them.

Minor comment in figure 3 , levels of Neu1 and neu4 show and increase following injury, but could you comment on why you think the levels of neu2 significantly goes down?

Thank you for your comment. The decrease in Neu2 expression suggests that it is expressed in mature neurons and other cells that would be decreased by injury. We have mentioned this possibility in the Results section as follows:

“The decrease in PSA intensity despite the decrease in Neu2 expression by brain injury may be due to the increased expression of Neu1 and Neu4, which are known to be involved in the cleavage of PSA. (Page 9, lines 255-257)”

“On the other hand, since Neu2 is reduced by brain injury, Neu2 may be expressed in cells that are reduced by injury, such as mature neurons. (Page 9, lines 266-268)”

5th Apr 2024

Dear Dr. Sawamoto,

Thank you for the submission of your revised manuscript to EMBO Molecular Medicine. I am pleased to inform you that we will be able to accept your manuscript pending the following final amendments:

- 1) Authors: Please define the corresponding author on the title page of the manuscript. We also note a discrepancy of author's name: Kouichi Fujimoto in the manuscript and Koichi Fujimoto in our submission system. Please correct.
- 2) Abstract: I have gone through your text and revised it. Please review it and amend as you see fit:

In the injured brain, new neurons produced from endogenous neural stem cells form chains and migrate to injured areas and contribute to the regeneration of lost neurons. However, this endogenous regenerative capacity of the brain has not yet been leveraged for the treatment of brain injury. Here, we show that in healthy brain chains of migrating new neurons maintain unexpectedly large non-adherent areas between neighboring cells, allowing for efficient migration. In instances of brain injury, neuraminidase reduces polysialic acid levels, which negatively regulates adhesion, leading to increased cell-cell adhesion and reduced migration efficiency in mice and primates. The administration of zanamivir, a neuraminidase inhibitor used for influenza treatment, promotes neuronal migration toward damaged regions, fosters neuronal regeneration, and facilitates functional recovery. Together, these findings shed light on a new mechanism governing efficient neuronal migration in the adult brain under physiological conditions, pinpoint the disruption of this mechanism during brain injury, and propose a promising therapeutic avenue for brain injury through drug repositioning.

- 3) In the main manuscript file, please do the following:

- Please address all comments suggested by our data editors listed below:

o Figure legends:

1. Please note that a separate 'Data Information' section is required in the legends of figures 1a-c, e; 2a-b, d-h, m-t; 3a-g", i-l; 4b-f, h, j-s; 5b-e, g-h, j-k; 6b-h; EV 1c-i'; EV 2b-h; EV 3a-h, k-l; EV 4b-f.
2. Please note that the figure EV 2a; EV 4d-f, does not contain any statistical parameter, kindly rectify the statistical test related information in the figure legend appropriately.
3. Please define the annotated p values ****/***/**/* in the legend of figure 1d, f; 2d-e, h, o-p, t; 3a-e, k-l; 4e-f, h, n, s, 5e, g-h, j-k; 6d-e, h; EV 1f-g; EV 2d, h; as appropriate.
4. Please indicate the statistical test used for data analysis in the legend of figure 5h.
5. Please note that information related to n is missing in the legend of figure 5h.
6. Although 'n' is provided, please describe the nature of entity for 'n' in the legends of figures 2d; 3e, l; 5e, g; EV 4d-f.
7. Please note that scale bar and its definition are missing for figures 2i-k.
8. Please note that the black arrowheads are not defined in the legend of figure 4o-r. This needs to be rectified.
9. Please note that the yellow arrows are not defined in the legend of figure 5b-d. This needs to be rectified.

- Author contributions: Please remove it from the manuscript and specify author contributions in our submission system. CRediT has replaced the traditional author contributions section because it offers a systematic machine-readable author contributions format that allows for more effective research assessment. You are encouraged to use the free text boxes beneath each contributing author's name to add specific details on the author's contribution. More information is available in our guide to authors:

<https://www.embopress.org/page/journal/17574684/authorguide#authorshipguidelines>

- Move data availability statement to the end of the "Methods".

- 4) Movies: Please remove movie legends from the main manuscript file and zip them with each corresponding movie file.
- 5) Appendix: Move all supplemental methods to the "Methods" section in the main manuscript.
- 6) Funding: Please make sure that information about all sources of funding are complete in both our submission system and in the manuscript. Currently, Bilateral Open Partnership Joint Research Projects, and Core-to-core Program "Neurogenesis Research & Innovation Center"), Cooperative Study Programs of National Institute for Physiological Sciences., Takeda Science Foundation [to K.S.], Nitto Foundation the Valencian Council for Innovation, Universities Science and Digital Society (PROMETEO/2019/075), and the Spanish Ministry of Science Innovation and Universities (PCI2018-093062) are missing in our submission system.

- 7) The Paper Explained: I have gone through your text and revised it. Please review it and amend as you see fit:

Problem:

Regenerative medicine is expected to be a fundamental treatment for brain injury. However, conventional methods are invasive, and the therapeutic effect is limited. For many years, this has remained a major challenge to the clinical application of regenerative medicine.

Results:

In this study, we show that the migration of new neurons produced from endogenous stem cells to the injured area is suppressed by neuraminidase, which is upregulated in the injured brain. Increased neuraminidase expression result in a decreased polysialic

acid (PSA) level, excessive cell-cell adhesion, and reduced migration efficiency. Inhibition of neuraminidase by clinically used neuraminidase inhibitors sustains PSA levels in new neurons and allows them to migrate to injured areas promoting neuronal regeneration and recovery of the brain function.

Impact:

We demonstrated that a minimally invasive and broadly effective treatment of brain injury can be achieved by drug administration and proposed a novel treatment method for brain injury. Furthermore, we demonstrated the potential of drug repositioning, in which drugs used to treat influenza are applied to the treatment of brain injury.

8) Synopsis:

- Synopsis text: I have gone through your text and revised it. Please review it and amend as you see fit:

There is currently no definitive treatment for brain injury. We have shown that inhibiting the cleavage of polysialic acid, responsible for maintaining space between new neurons produced from endogenous neural stem cells, can promote neuronal migration and regeneration. Additionally, we demonstrated the possibility to repurpose drugs, in clinical use for treatment of influenza, for the treatment of brain injury.

Bullet points:

1. Appropriate space is maintained in new neurons migrating collectively in the normal brain, whereas space is reduced, and cell adhesion is increased in new neurons migrating in the injured brain.
2. Brain injury induces a reduction in polysialic acid (PSA), which is important for the maintenance of space between new neurons, by increasing the expression of the neuraminidases.
3. Neuraminidase inhibitor treatment of injured brain maintained PSA level in new neurons, resulting in the promotion of neuronal migration, neuronal regeneration, and recovery of brain function.

9) For more information: This space should be used to list relevant web links for further consultation by our readers. Could you identify some relevant ones and provide such information as well? Some examples are patient associations, relevant databases, OMIM/proteins/genes links, author's websites, etc...

10) As part of the EMBO Publications transparent editorial process initiative (see our Editorial at <http://embomolmed.embopress.org/content/2/9/329>), EMBO Molecular Medicine will publish online a Review Process File (RPF) to accompany accepted manuscripts. This file will be published in conjunction with your paper and will include the anonymous referee reports, your point-by-point response and all pertinent correspondence relating to the manuscript. Let us know whether you agree with the publication of the RPF and as here, if you want to remove or not any figures from it prior to publication. Please note that the Authors checklist will be published at the end of the RPF.

11) Please provide a point-by-point letter INCLUDING my comments as well as the reviewer's reports and your detailed responses (as Word file).

I look forward to reading a new revised version of your manuscript as soon as possible.

Yours sincerely,

Zeljko Durdevic

*** Instructions to submit your revised manuscript ***

When submitting your revised manuscript, please

include:

1) a .docx formatted version of the manuscript text (including Figure legends and tables)

2) Separate figure files*

3) supplemental information as Expanded View and/or Appendix. Please carefully check the authors guidelines for formatting Expanded view and Appendix figures and tables at <https://www.embopress.org/page/journal/17574684/authorguide#expandedview>

4) a letter INCLUDING the reviewer's reports and your detailed responses to their comments (as Word file).

5) The paper explained: EMBO Molecular Medicine articles are accompanied by a summary of the articles to emphasize the major findings in the paper and their medical implications for the non-specialist reader. Please provide a draft summary of your article highlighting

6) For more information: There is space at the end of each article to list relevant web links for further consultation by our readers. Could you identify some relevant ones and provide such information as well? Some examples are patient associations, relevant databases, OMIM/proteins/genes links, author's websites, etc...

7) Author contributions: the contribution of every author must be detailed in a separate section.

8) EMBO Molecular Medicine now requires a complete author checklist (<https://www.embopress.org/page/journal/17574684/authorguide>) to be submitted with all revised manuscripts. Please use the checklist as guideline for the sort of information we need WITHIN the manuscript. The checklist should only be filled with page numbers where the information can be found. This is particularly important for animal reporting, antibody dilutions (missing) and exact values and n that should be indicated instead of a range.

9) Every published paper now includes a 'Synopsis' to further enhance discoverability. Synopses are displayed on the journal webpage and are freely accessible to all readers. They include a short stand first (maximum of 300 characters, including space) as well as 2-5 one sentence bullet points that summarise the paper. Please write the bullet points to summarise the key NEW findings. They should be designed to be complementary to the abstract - i.e. not repeat the same text. We encourage inclusion of key acronyms and quantitative information (maximum of 30 words / bullet point). Please use the passive voice. Please attach these in a separate file or send them by email, we will incorporate them accordingly.

You are also welcome to suggest a striking image or visual abstract to illustrate your article. If you do please provide a jpeg file 550 px-wide x 300-800px high.

10) A Conflict of Interest statement should be provided in the main text

11) Please note that we now mandate that all corresponding authors list an ORCID digital identifier. This takes <90 seconds to complete. We encourage all authors to supply an ORCID identifier, which will be linked to their name for unambiguous name identification.

Currently, our records indicate that the ORCID for your account is 0000-0003-1984-5129.

Please click the link below to modify this ORCID:
Link Not Available

Graphs 800-1,200 DPI
Photos 400-800 DPI
Colour (only CMYK) 300-400 DPI"

*Additional important information regarding figures and illustrations can be found at <https://bit.ly/EMBOPressFigurePreparationGuideline>. See also figure legend preparation guidelines: <https://www.embopress.org/page/journal/17574684/authorguide#figureformat>

***** Reviewer's comments *****

Referee #1 (Remarks for Author):

Thank you for an excellent revision. All my points have been clarified. The paper is now ready for publication.

The authors addressed the minor editorial issues.

19th Apr 2024

Dear Dr. Sawamoto,

We are pleased to inform you that your manuscript is accepted for publication and is now being sent to our publisher to be included in the next available issue of EMBO Molecular Medicine.
